# Proteome-wide analysis of phospho-regulated PDZ domain interactions

Gustav N Sundell[1], Roland Arnold[2,*], Muhammad Ali[1] (iD), Piangfan Naksukpaiboon[2], Julien Orts[3], Peter Güntert[3,4], Celestine N Chi[5,**] & Ylva Ivarsson[1,***] (iD)

## Abstract

A key function of reversible protein phosphorylation is to regulate protein–protein interactions, many of which involve short linear motifs (3–12 amino acids). Motif-based interactions are difficult to capture because of their often low-to-moderate affinities. Here, we describe phosphomimetic proteomic peptide-phage display, a powerful method for simultaneously finding motif-based interaction and pinpointing phosphorylation switches. We computationally designed an oligonucleotide library encoding human C-terminal peptides containing known or predicted Ser/Thr phosphosites and phosphomimetic variants thereof. We incorporated these oligonucleotides into a phage library and screened the PDZ (PSD-95/Dlg/ZO-1) domains of Scribble and DLG1 for interactions potentially enabled or disabled by ligand phosphorylation. We identified known and novel binders and characterized selected interactions through microscale thermophoresis, isothermal titration calorimetry, and NMR. We uncover site-specific phospho-regulation of PDZ domain interactions, provide a structural framework for how PDZ domains accomplish phosphopeptide binding, and discuss ligand phosphorylation as a switching mechanism of PDZ domain interactions. The approach is readily scalable and can be used to explore the potential phospho-regulation of motif-based interactions on a large scale.

**Keywords** PDZ domain; phage display; phosphorylation; protein–protein interaction; Scribble
**Subject Categories** Methods & Resources; Post-translational Modifications; Proteolysis & Proteomics
**Mol Syst Biol. (2018) 14: e8129**

## Introduction

Reversible protein phosphorylation is crucial for regulation of cellular processes and primarily occurs on Ser, Thr, and Tyr residues in eukaryotes (Seet *et al*, 2006). Phosphorylation may have different functional effects on the target protein, such as inducing conformational changes, altering cellular localization, or enabling or disabling interaction sites. Hundreds of thousands of such phosphosites have been identified in different cell lines and under different conditions (Olsen *et al*, 2006; Hornbeck *et al*, 2015). An unresolved question is which of these phosphosites are of functional relevance and not background noise caused by the off-target activity of kinases revealed by the high sensitivity in the mass spectrometry analysis. So far, only a minor fraction of identified phosphosites has been linked to functional effects, and only 35% of phosphosites are evolutionarily conserved, which would support that they play non-redundant functional roles (Landry *et al*, 2009). Sifting functional from non-functional phosphosites by experimental approaches is thus of fundamental importance in order to better understand the function of the phosphoproteome.

Phosphosites are often located in the intrinsically disordered regions of proteins, which in eukaryotes are estimated to cover 30–40% of the populated protein sequence space (Ward *et al*, 2004). These regions are enriched in short linear motifs (SLiMs, 3–12 amino acids) that are recognized by modular domains (Van Roey *et al*, 2014). Discrete phosphorylation events may regulate and tune the strength of interactions. Phosphopeptide binding domains, such as the 14-3-3 proteins, serve as readers of this phosphorylation code (Yaffe *et al*, 1997; Reinhardt & Yaffe, 2013). Deciphering the phosphorylation code by linking the reader domains to their preferred binding phosphosites is a major challenge. Adding to the complexity, phosphorylation of a SLiM may have a switch-like effect, making the interaction stronger (enabling) with a given domain, while weakening interactions (disabling) with other domains (Van Roey *et al*, 2013), in other worlds regulating interactions. Such a switch-like mechanism was recently found to regulate interactions

1   Department of Chemistry − BMC, Uppsala University, Uppsala, Sweden
2   Institute of Cancer and Genomic Sciences, College of Medical and Dental Sciences, University of Birmingham, Edgbaston, Birmingham, UK
3   Laboratory of Physical Chemistry, ETH Zürich, Zürich, Switzerland
4   Institute of Biophysical Chemistry, Goethe University, Frankfurt am Main, Germany
5   Department of Medical Biochemistry and Microbiology, Uppsala University, Uppsala, Sweden
    *Corresponding author. Tel: +44 7936624996; E-mail: r.arnold.2@bham.ac.uk
    **Corresponding author. Tel: +46 18 471 4557; E-mail: chi.celestine@imbim.uu.se
    ***Corresponding author. Tel: +46 18 471 40 38; E-mail: ylva.ivarsson@kemi.uu.se

of the C-terminal region of PRTM5 with 14-3-3 proteins and PDZ (PSD-95/Discs-large/ZO-1) domains. Phosphorylation of the C-terminal tail of PRTM5 enables 14-3-3 interactions, while disabling PDZ domain interactions (Espejo *et al*, 2017). Although less common, phosphorylation switches also operate on the interaction interfaces of folded proteins, hundreds of which were suggested in recent analysis (Betts *et al*, 2017).

There is a paucity of information linking phosphorylation enabled/disabled SLiMs to their binding partners, in part due to a lack of suitable experimental methods. Although affinity purification coupled to mass spectrometry (AP-MS) can be used to obtain information on phosphorylation states and interactions dynamically, the approach does not provide information with resolution at the level of the binding sites. In addition, interactions that rely on SLiMs are often elusive to AP-MS as they typically are of low-to-medium affinities, and exhibit rapid association/dissociation kinetics. Phosphopeptide libraries can be used to establish preferred binding motifs of phosphopeptide binding domains, but the number of sequences presented in these experiments is typically limited (Yaffe & Smerdon, 2004). In addition, Grossman and co-workers recently showed the use of yeast-two-hybrid (Y2H) for capturing phospho-Tyr-dependent interactions (Grossmann *et al*, 2015), but phospho-dependent interactions are otherwise typically not captured by Y2H. Thus, there is an urge for novel large-scale methods for charting phosphorylation-dependent SLiM-based interactions.

Here, we describe a novel large-scale approach termed phosphomimetic proteomic peptide-phage display (phosphomimetic ProP-PD) developed to identify SLiM-based interactions that are regulated by Ser/Thr phosphorylation. Phosphomimetic ProP-PD is a further development of ProP-PD, which is a dedicated method for the large-scale identification of SLiM-based interactions (Ivarsson *et al*, 2014; Davey *et al*, 2017; Wu *et al*, 2017). We demonstrate that phosphomimetic ProP-PD is a straightforward approach for finding ligands of SLiM-binding domains and for simultaneously pinpointing Ser/Thr phosphorylation events with potential to enable or disable interactions. We showcase the approach by identifying phospho-regulated interactions of PDZ (PSD-95/Dlg/ZO-1) domains, which are among the most frequent interaction modules in eukaryote proteomes with about 267 instances in over human 150 proteins (Luck *et al*, 2011; Ivarsson, 2012). PDZ domains are well known for binding to C-terminal peptides with typical PDZ binding motifs (PDZbms; class I binding motif S/T-x-Φ-coo-, class II Φ-x-Φ-coo-, and class III D/E-x-Φ-coo-, where x is any amino acid and Φ is a hydrophobic amino acid), but also interact with internal PDZbms and phospholipids (Ivarsson, 2012). The last amino acid in the PDZbm is denoted p0, the upstream residue is denoted p-1, and so on. Although the PDZbms are the crucial determinants for binding, residues upstream contribute to the interactions (Tonikian *et al*, 2008). Based on a limited literature on phospho-regulation of PDZ domain interactions, phosphorylation of the Ser/Thr at p-2 of the class I PDZbm typically disables PDZ interactions (Cohen *et al*, 1996; Chetkovich *et al*, 2002; Choi *et al*, 2002; Hu *et al*, 2002; Tanemoto *et al*, 2002; Toto *et al*, 2017), while phosphorylation of other positions may either enable or disable interactions (Clairfeuille *et al*, 2016).

In this study, we first create a phosphomimetic ProP-PD library displaying C-terminal regions of the human proteome that contain

known or predicted phosphorylation sites, and the phosphomimetic mutants of these peptides. Second, we use this library in selections against the six PDZ domains of human Scribble and DLG1, well-characterized PDZ proteins with crucial roles in the postsynaptic density of excitatory neuronal synapses and in the establishment and maintenance of epithelial cell polarity (Humbert *et al*, 2008; Feng & Zhang, 2009). Third, we successfully identify known and novel interactions, with potential to be regulated by Ser/Thr phosphorylation, and uncover that Scribble PDZ interactions are enabled by p-3 phosphorylation. Finally, we provide the structural basis of such phosphopeptide binding, expand the analysis to an additional nine domains, and discuss the role of Ser/Thr phosphorylation as a switching mechanism of PDZ domain interactions. In this proof-of-concept study, we thus demonstrate that phosphomimetic ProP-PD is a powerful approach to identify SLiM-based interactions in the human proteome that are regulated by Ser/Thr phosphorylation. The approach is readily scalable and can be used to explore the potential phospho-regulation of human protein–protein interactions on a larger scale.

## Results

### Construction of a phosphomimetic ProP-PD library

To design a specific phage display library for phosphomimetic ProP-PD, we scanned the human proteome for C-terminal regions containing known or predicted Ser/Thr phosphorylation sites based on phosphorylation databases and *in silico* predictions (see Materials and Methods). We identified 4,828 unique C-terminal peptides of human full-length proteins that contain one or multiple potential Ser/Thr phosphosites in their last nine amino acids. We designed the phosphomimetic ProP-PD library to comprehensively contain wild-type sequences and phosphomimetic mutants (Ser/Thr to Glu) thereof (7,626 phosphomimetic sequences, including combinations of multiple phosphorylation sites in one C-terminus), in order to allow the detection of interactions that are enabled or disabled by phosphomimetic mutations. The library design contains 24% of all non-redundant C-terminal peptides of human full-length proteins reported in the annotated section of SwissProt/UniProt (Table EV1). Of these sequences, 8.2% contain class I PDZ binding motifs, 6.2% contain class II binding motifs, and 3.4% match class III binding motifs. In case of the class I-containing peptides, 40.8% of the phosphorylation sites are at p-2 and are expected to disable PDZbm-dependent interactions.

Oligonucleotides coding for the wild-type and phosphomimetic (Ser/Thr to Glu) peptides flanked by annealing sequences (Table EV2) were obtained as a custom oligonucleotide library and incorporated into a phagemid vector, thereby creating C-terminal fusion proteins of the M13 major coat protein p8, previously engineered for C-terminal display (Fuh *et al*, 2000; Fig 1). We confirmed 94.6% of the designed sequences by NGS and the vast majority of reads matched the library design. 62% of raw library sequences were of right length, and of those, non-synonymous mutations were present in 14.6% of the sequences. There were no major sequence biases in the constructed phage library (Fig EV1). The library coverage and quality were thus considered satisfactory.

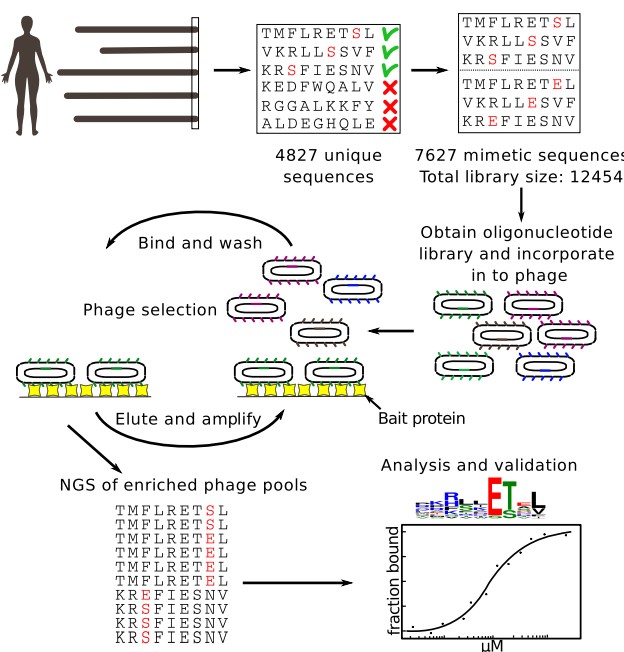

**Figure 1. Schematics of the phosphomimetic ProP-PD approach and validations.**

We designed a phosphomimetic ProP-PD library to display the C-terminal regions (nine amino acids) of human proteins containing known or putative Ser/Thr phosphorylation sites (4,827 peptides), and all phosphomimetic (Ser/Thr to Glu) variants thereof (7,626 peptides). Oligonucleotides encoding the sequences were incorporated into a phagemid designed for the display of peptides fused to the C-terminus of the major coat protein P8 of the M13 filamentous phage. The library was used in binding selections against immobilized PDZ domains. Binding-enriched phage pools were analyzed by NGS, followed by data analysis and validations.

## Phosphomimetic ProP-PD selections against the PDZ domains of Scribble and DLG1

We tested the performance of the phosphomimetic ProP-PD library through selections against the immobilized PDZ domains of human Scribble (PDZ1, PDZ2, and PDZ3) and DLG1 (PDZ1, PDZ2, and PDZ3). Of these proteins, Scribble has previously been shown to interact with the p-1 phosphorylated PDZbm of MCC (colorectal mutant cancer protein; RPHTNETpSL-coo-; Pangon *et al*, 2012). In contrast, phosphorylation of the p-2 position has been shown to disable interaction with DLG1 PDZ2 (Adey *et al*, 2000). The experiment was thus designed to capture interactions that were enabled or disabled by phosphomimetic mutations. The selections were successful as judged by phage pool enzyme-linked immunosorbent assay (ELISA) and saturated after three rounds of selection. Binding-enriched phage pools were barcoded and analyzed by NGS, which resulted in a list of binding peptides ranked by their occurrence in the NGS results. Although we found increased sensitivity of a peptide being selected during the phage display if it is well represented in the naïve phage libraries (see boxplot figures of initial read support versus being selected or not selected; Fig EV1), there is no correlation of the selection strength with representation in the naïve library in any of the experiments reflecting the strong selection during phage display (Fig EV1).

High-confidence sets of ligands (Fig 2A; Table EV3) were obtained by assigning cutoff values that filtered out non-specifically retained peptides lacking typical PDZbms. For each domain, we generated position weight matrices (PWMs) based on the ProP-PD (Fig 2B). All domains are class I binding PDZ domains, and the PWMs are in good agreement with previous studies (Tonikian *et al*, 2008; Ivarsson *et al*, 2014; Karlsson *et al*, 2016).

### Analysis of the effects of phosphomimetic mutations

We systematically evaluated the effects of phosphomimetic mutations at distinct positions of the PDZbms by analyzing the NGS counts received for each peptide pair (wild-type and phosphomimetic; Fig 2A). Essentially, we calculated the ratio between the NGS counts of a given phosphomimetic peptide and the sum of NGS counts of the phosphomimetic peptide and its wild type (Table EV3). For comparison between replicate experiments, we used the fraction of counts, rather than the raw counts, as the total amount of NGS reads differed between the sequencing rounds. We then calculated the average ratios for each position of the PDZbm. This resulted in a score in the range between 0 and 1 for each peptide position, where 0 indicates that the selection was dominated by wild-type peptides and suggests that phosphomimetic mutations at this position disable interactions. A score of 1 suggests that phosphomimetic mutations at the given position enable interactions. The scores for the different domains and peptide positions are summarized in matrices (Fig 2C). Inspection of the NGS frequency of the wild-type and mutant peptides and the matrices revealed common and distinct features. Among the shared features is that phosphomimetic mutations at p-2 disable interactions with all PDZ domains. Phosphorylation of this position is thus expected to serve as an "off-switch" for class I PDZ-mediated interactions. Interestingly, the analysis suggests that phosphomimetic mutations of the p-3 position enable interactions, in particular for Scribble PDZ1 and Scribble PDZ3, for which the differences are statistically significant (Fig 2C, Table EV4). Phosphomimetic mutations of the upstream positions −5 to −6 have instead negative effects on the interactions. Phosphorylation may thus enable or disable PDZ-mediated interactions in a site-specific manner.

### Microscale thermophoresis affinity measurements confirm phosphorylation switches

To explore the extent to which the phosphomimetic ProP-PD results translate into affinity differences between unphosphorylated and phosphorylated peptides, we focused on Scribble PDZ1 ligands. We selected three distinct sets of peptides (unphosphorylated, phosphomimetic mutant, and phosphorylated) for affinity determination by microscale thermophoresis (MST; Fig 3A). The peptides were chosen in order to validate the effects of the putative enabling phosphorylation switches at the p-1 (MCC; HTNET**S**L-coo-) and p-3 (RPS6KA2, RLT**S**TRL-coo-) positions, and the disabling effect of the upstream p-6 position (TANC1, KR**S**FIESNV-coo-). Scribble PDZ1 was titrated against FITC-labeled wild-type, phosphomimetic, and phosphorylated peptides. All peptides bound to Scribble PDZ1 with micromolar affinities (0.6–101 μM $K_D$ values; Fig 3A). Consistent with the phosphomimetic ProP-PD results, Scribble PDZ1 has a higher affinity (3×) for the p-1 phosphorylated MCC peptide and the

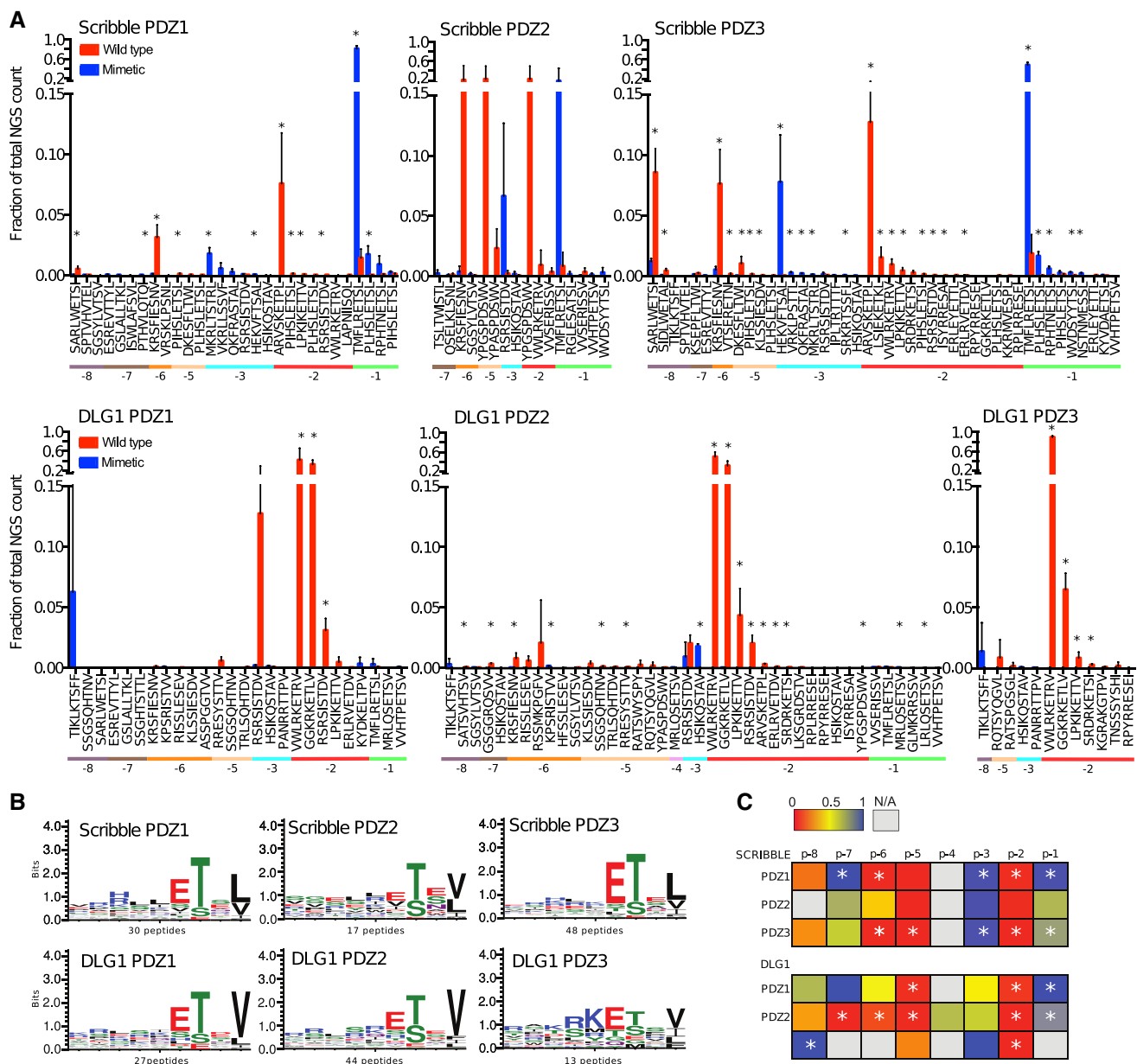

**Figure 2.  Analysis of the phosphomimetic ProP-PD selection results.**

A    Fractions of the sequencing counts for individual peptides (wild-type peptides indicated by red bars and phosphomimetic variants by blue bars) from NGS analysis of binding-enriched phage pools from selections against the PDZ domains of Scribble and DLG1. Peptides are sorted based on the site of the phosphomimetic mutations (corresponding to known or putative phosphosites). Standard deviations are indicated ($n = 3$ for all cases except for Scribble PDZ2 where $n = 2$). Peptide pairs for which there are significant differences between the fractions of NGS counts for wild-type and phosphomimetic peptides are indicated with * (multiple *t*-tests, significance level 0.05). Peptides with multiple phosphorylation sites have been omitted from this analysis for clarity. For further details, see Table EV3.

B    Position weight matrices (PWMs, WebLogo3) representing the phosphomimetic ProP-PD selections data for the PDZ domains of Scribble and DLG1. The numbers of peptides used for the analysis are indicated.

C    Scoring matrices of the effects of phosphomimetic mutations on the phosphomimetic ProP-PD results of the PDZ domains of Scribble and DLG1. Scores are calculated from the ratios between NGS counts of the phosphomimetic peptides and the sum of the NGS counts of the wild-type and phosphomimetic peptides. A score of 0 (red) indicates that the selection was dominated by wild-type peptides, and a score of 1 (blue) indicates that the selection was dominated by peptides with phosphomimetic mutations at the given position ($n = 3$ for all cases except for Scribble PDZ2 where $n = 2$). Positions marked with * have a score that is significantly different from the score 0.5 of a neutral effect as determined using multiple *t*-tests and correcting for multiple comparisons using a false discovery rate of 2.5%. The position for which only wild-type or phosphomimetic peptides are represented in the data set could not be subjected to this test.

p-3 phosphorylated RPS6KA2 (5×), as compared to their unphosphorylated counterparts. The opposite is true for the TANC1 peptide (2× lower affinity for phosphopeptide; Fig 3A). Thus, there is a good

qualitative agreement between the phosphomimetic ProP-PD data and the affinity differences between phosphorylated and unphosphorylated ligands as determined by MST for this domain. The

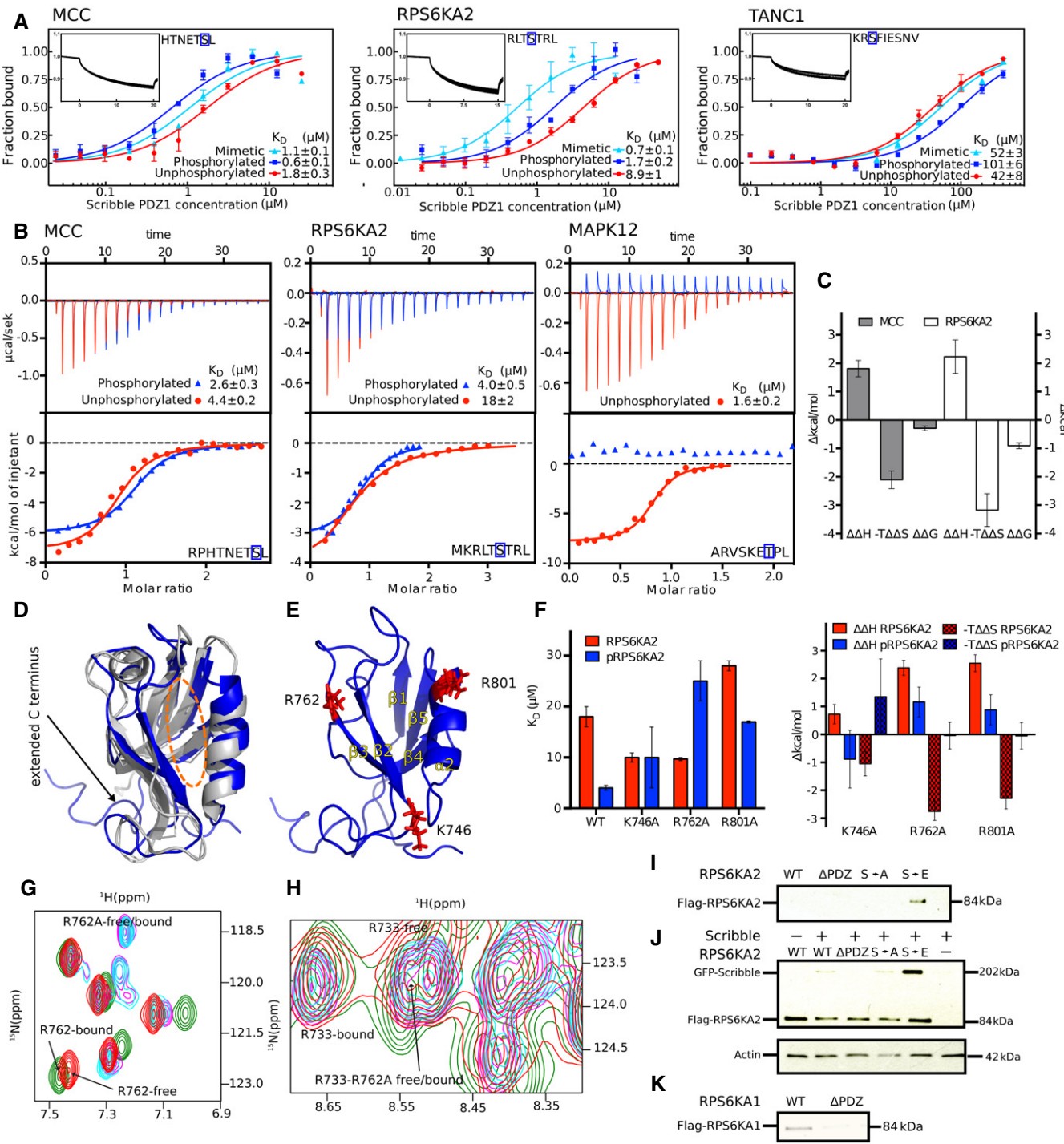

**Figure 3.**

effects of the phosphomimetic mutations are qualitatively, but not quantitatively, the same as the effects of ligand phosphorylation, which is expected due to the distinct properties of Glu in comparison with phospho-Ser/Thr.

To evaluate the effects of ligand phosphorylation on DLG1 PDZ binding, we determined the affinities for the same set of peptides with DLG1 PDZ2 (Fig EV2; Table EV5), and found them to be in the μM range (2.7–95 μM). MST measurements with the p-1 modified

MCC peptides revealed that phosphorylation of this site in this ligand has a disabling effect on DLG1 PDZ2 interactions (9×; $K_D^{MCC}$ 6.4 μM; $K_D^{pMCC}$ 57 μM). This seemingly contradicts the result reported in the position-specific matric, but importantly, the MCC ligand was not identified as a DLG1 PDZ2 in the phage selection, and it is possible that the results may not be generally extended to other ligands, as both domains and ligands are flexible and the results of phosphorylation may depend on the combination of

**Figure 3. Scribble PDZ1 preferentially interacts with p-1 and p-3 phosphorylated ligands as shown by affinity determinations (MST and ITC), NMR structure, mutational analysis, and GST-pulldowns and co-immunoprecipitation experiments.**

A   MST affinity measurements using FITC-labeled peptides (unphosphorylated, phosphorylated, and phosphomimetic variants) of MCC (p-1), RPS6KA2 (p-3), and TANC1 (p-6). A fixed peptide concentration (25–50 nM) was titrated with varying concentrations of Scribble PDZ1. $K_D$ values were determined using thermophoresis and T-Jump signal for data analysis (n = 3; error bars represent SD). Statistical assessment using ordinary one-way ANOVA for multiple comparisons confirmed significant differences ($P \leq 0.001$) between the affinities for the phosphorylated peptides compared to the unphosphorylated peptides of MCC, RPS6KA2, and TANC1. For the comparison between the wild-type and phosphomimetic peptides, there were significant differences for the peptides of MCC ($P \leq 0.01$) and RPS6KA2 ($P \leq 0.001$), but not for TANC1. Insets show representative titrations.

B   Representative ITC titrations of Scribble PDZ1 with unphosphorylated and phosphorylated peptides of MCC, RPS6KA2, and MAPK12 (n = 3). Statistical assessment using multiple *t*-tests corrected using the Holm–Sidak method for multiple comparisons show a significant difference between the phosphorylated and unphosphorylated MCC ($P = 0.00098$) and RPS6KA2 ($P = 0.0003$) peptides.

C   Differences between the thermodynamic parameters of Scribble PDZ1 when binding unphosphorylated or phosphorylated ligands (MCC and RPS6KA2) as determined in (B) (error bars represent SD, n = 3).

D   Overlay of the previously published unliganded structure of Scribble PDZ1 (gray; PDB code 1X5Q) and the here determined NMR structure (blue; PDB code 6ESP) of the protein bound to MKRLTpSTRL-coo- (peptide not shown). The canonical PDZ binding grove is indicated by a dotted orange line.

E   Structure of the phosphopeptide-bound Scribble PDZ1 showing three positively charged residues (K746, R762, and R801) surrounding the peptide binding pocket.

F   Left panel: Equilibrium binding constants for the binding between Scribble PDZ1 wild type and mutants (K746A, R762A, and R801A) and RPS6KA2 (unphosphorylated and phosphorylated) as determined through ITC titrations (n = 3). Right panel: Changes in the thermodynamic parameters upon mutation ($\Delta\Delta H^{mutation}$ and $-T\Delta\Delta S^{mutation}$) as determined through ITC (error bars represent SD, n = 3).

G   Section of the $^{15}$N-$^{1}$H HSQC spectra of ligand-free wild-type Scribble PDZ1 (red) and the protein bound to the phospho-RPS6KA2 peptide (green), together with the spectra of the ligand-free R762A mutant (magenta) and the mutant bound to phospho-RPS6KA2 (cyan). Note that residue 762 experiences chemical shift perturbation only in the wild-type Scribble PDZ1.

H   Slice of the $^{15}$N-$^{1}$H HSQC spectra showing the R733 or R762A residue of the wild-type or mutant proteins. R733 is critical for mediating the carboxylate at the C-terminus, and its position does not change in the wild-type-bound or the R762A-bound form, showing that the carboxylate is still making the important interaction.

I   GST-pulldown of full-length Flag-tagged RPS6KA2 wild type and mutants (S730A and S730E) and a truncated version (S730Δ) over-expressed in HEK293T cells with GST-tagged Scribble PDZ1. Detection was performed using an anti-Flag antibody.

J   Co-IPs of GFP-tagged full-length Scribble with Flag-tagged full-length RPS6KA2 constructs as indicated. Detection was performed using an anti-Flag antibody and an anti-GFP antibody.

K   GST-pulldown of full-length Flag-tagged RPS6KA1 wild type and a truncated version of the protein (S732Δ). Detection was performed using an anti-Flag antibody.

Source data are available online for this figure.

---

amino acids at other ligand positions (e.g., the MCC peptide has a L at p0 instead of the V which is preferred by DLG1 PDZ2). For the p-3 modified RPS6KA2 peptide (Fig EV2), phosphomimetic mutation has a minor enabling effect on the interaction (1.3×). Both phosphorylation and phosphomimetic mutations of the p-6 position of the TANC1 peptide have minor disabling effects on DLG1 PDZ 2 binding (2.9× and 2.4×, respectively), consistent with the phosphomimetic ProP-PD results.

### Isothermal titration calorimetry uncovers entropic–enthalpic compensation

To further confirm the affinity differences determined for Scribble PDZ1, and to gain insights into the thermodynamics of the phosphopeptide interactions, we performed isothermal calorimetric (ITC) affinity measurements of Scribble PDZ1 binding to the unphosphorylated and phosphorylated peptides of MCC (p-1 RPHTNET**S**L-coo-) and RPS6KA2 (p-3 MKRLT**S**TRL-coo-; Fig 3B). In addition, we tested the effect of p-2 phosphorylation using unphosphorylated and phosphorylated MAPK12 peptides (ARVS-KET**P**L-coo-). Consistent with the phosphomimetic ProP-PD and MST results, phosphorylation of MCC at p-1 has a minor enabling effect (2× lower $K_D$) on the interactions, and phosphorylation of RPS6KA2 p-3 has an enabling effect (5×). Phosphorylation at p-2 of MAPK12 disables the interaction, to the extent that no interaction was observed within the concentration range used (25–2,200 μM), in line with the disabling effect of p-2 phosphorylation suggested by the phage display.

The differences in Gibbs free energy of binding ($\Delta\Delta G^{phos} = \Delta G^{phos} - \Delta G^{unphos}$) between phosphorylated and unphosphorylated peptides were modest, -0.29 kcal/mol for MCC and -0.89 kcal/mol for RPS6KA2 (Fig 3C). However, we note an entropy–enthalpy compensation effect between the binding of the unphosphorylated–phosphorylated peptide pairs. Moreover, the interaction between Scribble PDZ1 and RPS6KA2 (but not MCC) is associated with an increase in entropy ($-T\Delta S^{RPS6KA2} = -1.6$ kcal/mol and $-T\Delta S^{phosRPS6KA2} = -4.8$ kcal/mol). This may indicate that the PDZ domain becomes less rigid upon binding the peptides and/or that they have pre-formed structures that are broken upon binding, or that peptide binding is associated with the release of water. Taken together, affinity measurements through MST and ITC confirm that phosphomimetic ProP-PD can be used to identify interactions and to pinpoint putative phosphorylation switches, and the thermodynamic parameters reveal that the effects of phosphorylation on ligand binding are more complex than revealed by the modest affinity changes.

### NMR structural analysis of Scribble PDZ1 interactions with unphosphorylated and phosphorylated ligands

To further explore the novel interactions that we identified by phosphomimetic ProP-PD and confirmed through MST and ITC, we determined the NMR structure of the Scribble PDZ1 domain when bound to the phosphorylated RPS6KA2 (p-3) peptide. We also performed $^{15}$N-$^{1}$H heteronuclear single-quantum correlation spectrum (HSQC) titrations of the RPS6KA2 and MCC peptides (phosphorylated and unphosphorylated). The bound Scribble PDZ1 structure has the typical PDZ fold, containing six β-strands and two α-helices arranged in a sandwich pattern (Fig 3D). The structure was overall similar to the free protein except for the noticeable opened binding pocket and the presence of an extended C-terminal

region comprising residues 816–829, which is packed onto the β2/β3 loop. This extended C-terminal region seems to suggest a role of extra structural elements outside the canonical PDZ fold. This region was not determined in the free Scribble PDZ1 domain, perhaps because the sequence used for the structure determination was shortened at both the C- and N-termini (PDB codes 1X5Q and 2W4F). However, we did not see any $^{15}$N-$^{1}$H chemical shift difference in the HSQC spectra of the amino acid residues comprising this region between the free and bound proteins. This suggests that the extra C-terminal structure is likely present in the free protein too and might function as an extra scaffold for stability. From the Scribble PDZ1 HSQC titration spectra, we observed chemical shift perturbations for both the RPS6KA2 and MCC peptides (phosphorylated and unphosphorylated), further indicating that all four peptides interact with Scribble PDZ1. The strength of the interactions could not be determined from these experiments as only two points of titrations (free and bound) were performed. Nevertheless, it gave atomic information of the interaction being probed (see below).

The NMR structure was determined with labeled protein and unlabeled, phosphorylated RPS6KA2 peptide (Table EV6). As such, only the protein was visible. However, we observed from the structure that K746, R762, and R801 are all surface-exposed and around the canonical binding pocket (Figs 3E and EV2). In addition, the $^{15}$N-$^{1}$H titration experiments revealed that these residues experienced characteristic shift changes. We hypothesized that one or a combination of these positively charged residues are responsible for stabilizing the extra charges brought in by the phosphate group at position p-3. We therefore introduced single-point mutations replacing the basic residues with neutral Ala (K746A, R762A, and R801A) and performed ITC experiments as above for phosphorylated and unphosphorylated RPS6KA2 peptides. Each of the three substitutions conferred reduced affinities for the phosphopeptide (Fig 3F), with the R762A mutation having the most pronounced effect on the affinity for phosphopeptide (6× weaker affinity). The R801A mutation also caused a reduced affinity for the unphosphorylated ligand and thus affected peptide binding in general. In contrast, the K746A and R762A mutations resulted in stronger affinities for the unphosphorylated ligand; in case of the R762A mutation, phosphorylation has a disabling effect on ligand binding instead of enabling (Fig 3F). The ITC results revealed that the effect of the R762A mutation on phosphopeptide binding is caused by a change in the enthalpic contribution to binding, and the residue also points into the cavity likely to be occupied by the phosphate group (Figs 3E and EV3). The effect of K746A is a bit unclear, but perhaps it plays a role in steering the charged residues at −6 and −7 of the peptide. This observation comes from the HSQC titration experiment since both the phosphorylated and unphosphorylated peptides from MCC and RPS6KA2 had huge chemical shift perturbation on K746, which is located at the end of the binding pocket. R801 on the other hand might play a steering role of both the C-terminal carboxylate and the phosphate group that are located in its vicinity.

To further explore the effect of the R762A mutation, we performed $^{1}$H-$^{15}$N HSQC titrations of Scribble PDZ1 R762A and unphosphorylated and phosphorylated RPS6KA2. The titrations revealed that the large conformational change exhibited by wild-type Scribble PDZ1 upon binding phospho-RPS6KA2 is lost in the R762A mutant (Fig 3G). In contrast, the position of R733, which is

critical for mediating interactions with the carboxylate at the C-terminus, does not change in the wild-type-bound or the R762A-bound form, showing that the carboxylate of the peptide is still making the important interaction (Fig 3H), supporting that the domain is still functional in binding unphosphorylated and phosphorylated RPS6KA2.

Our structural model, titration, and mutagenesis analysis provide mechanistic insights into how Scribble PDZ1 binds the p-3 phosphorylated RPS6KA2 peptide, and establish that R762 can be considered a gate keeper residue that provides a specificity for the p-3 phosphorylated ligand. Analysis of the sign of the NMR chemical shift differences can further be used to explore the molecular basis of the differences between binding the unphosphorylated and phosphorylated ligands (Sundell *et al*, 2018).

## GST-pulldown and co-immunoprecipitation of Scribble and RSK6A2

We next aimed to verify whether the identified interactions and putative phosphorylation switches are functional in the context of the full-length proteins focusing on the interaction between Scribble and RPS6KA2. We performed GST-pulldown experiments between GST-tagged Scribble PDZ1 and Flag-tagged RPS6KA2 wild type (WT), phosphomimetic mutant (S730E), Ala mutant (S730A), and a deletion construct where the three last amino acids had been truncated (RPS6KA2Δ). We successfully confirmed an interaction with RPS6KA2 S730E, but not with the other constructs, supporting that the phosphomimetic mutation is required for interaction (Fig 3I). In addition, we performed co-immunoprecipitations (IPs) of transiently expressed GFP-tagged full-length Scribble and the Flag-tagged RPS6KA2 constructs and confirmed that full-length Scribble only shows strong co-IP with the phosphomimetic RPS6KA2 S730E (Fig 3J). The results thus support that phosphomimetic ProP-PD can identify putative phosphorylation-dependent interactions that are functional in the context of the full-length proteins.

Although RPS6KA2 S730 is predicted to be autophosphorylated, there is so far no experimental support for its phosphorylation. In contrast, the corresponding autophosphorylation site (S732) in the C-terminal tail (VRKLPSTTL-coo-) of the homologous RPS6KA1 is a confirmed phosphosite (Dalby *et al*, 1998). RPS6KA1 was also found as a Scribble PDZ1 ligand in phosphomimetic ProP-PD (Fig 2, Table EV3). We therefore determined the affinities of unphosphorylated and phosphorylated RPS6KA1 peptide for Scribble PDZ1 through MST using FITC-labeled peptides, resulting in a $K_D$ of 0.39 μM for the phosphorylated RPS6KA1 and 1 μM for the unphosphorylated peptide (Fig EV2, Table EV5). The results thus confirm that Scribble preferentially interacts with p-3 phosphorylated RPS6KA family members. The RPS6KA1 interaction was further confirmed through GST-pulldown of Flag-tagged RPS6KA1 (Fig 3K). In this case, the interaction was confirmed in the absence of a phosphomimetic mutation, which is not surprising given the high affinity of the interaction with the wild-type RPS6KA1 peptide. However, phosphorylation of RPS6KA1 p-3 may serve a switching function as the PDZbm overlaps with an ERK2 binding site, and phosphorylation of p-3 reduces the affinity for ERK2 binding (Roux *et al*, 2003), while increasing the affinity for Scribble PDZ1.

## Comparison between phosphomimetic ProP-PD results and known ligands

Gene ontology (GO) term enrichment analysis of the identified ligands (GOrilla, using the library design as background) revealed enrichment of GO terms related to Hippo signaling (Scribble) and excitatory chemical synaptic transmission (DLG1), and the ligand sets thus appear enriched in functionally relevant interactions (Table EV7). We compared the ligands identified through phosphomimetic ProP-PD with the known interactions listed in the HIPPIE database as of June 2018 (Alanis-Lobato *et al*, 2017; Fig 4A). HIPPIE incorporates information from databases such as IntAct (Orchard *et al*, 2014) and BioGRID (Chatr-Aryamontri *et al*, 2015) and contains already our previously generated ProP-PD results. Of the Scribble interacting proteins identified in this study, 25% are reported as Scribble ligands in the HIPPIE database, of which the evidences in 69% of cases are from our earlier study (Ivarsson *et al*, 2014). Among the interactions found through phosphomimetic ProP-PD, we note the interaction between Scribble and the Hippo pathway protein WWTR1 (KSEPFLTWL-coo-), which, alongside YAP1, functions as a transcriptional co-activator downstream of the Hippo pathway (Moroishi *et al*, 2015). Interestingly, our results also suggest that Scribble interacts with YAP1 (DKESFLTWL-coo-), which provides an additional link to the Hippo signaling pathway. In addition to the ligands reported in HIPPIE, additional literature support was found for two of the interactions that were previously predicted and validated by Luck *et al* (2011). For DLG1, 32.5% of the ligands identified through phosphomimetic ProP-PD were reported in the HIPPIE database, of which 62.5% were also found in our pilot ProP-PD study. Additional literature support was found for eight of the DLG1 ligands (Table EV3), among which the interaction with APC is of importance for regulating cell cycle progression (Matsumine *et al*, 1996).

Most of the ligands previously identified as binders through ProP-PD but not identified through phosphomimetic ProP-PD were missing from the novel library design as they did not contain Ser/Thr phosphorylation sites (89% for Scribble; 86% for DLG1). The overlap between ProP-PD and phosphomimetic ProP-PD demonstrate that the method robustly reports on overlapping sets of ligands despite differences in library designs such as different peptide length. In addition, phosphomimetic ProP-PD allows the identification of novel interactions potentially enabled by phosphorylation (47% of the novel Scribble ligands, 26% of the novel DLG1 ligands), thereby complementing protein–protein (PPI) data sets generated by other methods.

## Network analysis of ProP-PD results

As confirmed through biophysical experiments, phosphomimetic ProP-PD provides information on both ligands and phosphorylation switches. We visualized the multilevel information obtained through phosphomimetic ProP-PD in a PPI network where we included the ligands for which we found additional literature support and/or that share GO terms with the bait proteins that are unlikely to occur by chance (*P* < 0.01; Fig 4B). In this network, we indicate for which PDZ domain the interaction was identified (edge color), if the phosphomimetic ProP-PD results suggest the interaction to be enabled or disabled by phosphomimetic mutation (above/

below dotted line, respectively), and the position of the known or putative phosphorylation site (node color).

To present the novel Scribble and DLG1 interactions in a more functional context, we created a STRING network of the bait proteins and their ligands, where we included additional proteins if they have high-confidence connections to the bait proteins or to the identified ligands. As this network contains too many connections to be visualized in a meaningful way, we filtered the network for proteins within KEGG pathways selected based on STRING-based enrichment analysis (Szklarczyk *et al*, 2015). In addition to the previously mentioned Hippo signaling pathway, the network is enriched in connections to Wnt signaling, oxytocin signaling pathway, and neuroactive ligand receptor interactions (Fig 4C).

## Analysis from the perspective of the kinases

We further evaluated the phosphomimetic ProP-PD data from the perspective of the kinase that are known or predicted to phosphorylate the identified ligands (Table EV8), using the NetworKIN algorithm, which takes into account binding motif scores as well as phylogenetic relationships between kinases and binders and network proximity in a functional interaction network (Linding *et al*, 2007). From this analysis, we found that Scribble PDZ domains interact with potential RSK, type II receptor Ser/Thr kinase (ACTR2_ACTR2B_TGFbR2_group), and GSK3 substrates more frequently than expected when compared to the frequencies of these kinases' substrates in the library design (*P*-values 0.0002 and 0.09 for the other two, respectively, Fisher's one-sided exact test, BH correction for multiple testing), although substrates of other kinases are also represented. Based on the phosphomimetic ProP-PD results, RSK phosphorylation generally enables Scribble interactions, as is the case of the autophosphorylation of RPS6KAs, while phosphorylation by the ACTR2_ACTR2B_TGFbR2_group disables interactions. For example, based on the phosphomimetic ProP-PD results and the NetworKIN predictions, Scribble's interaction with YAP1 is proposed to be negatively regulated by p-5 phosphorylation by TGFbR2. The YAP1 p-5 phosphorylation site has been confirmed through phosphoproteomics (Kettenbach *et al*, 2011). Thus, through a combination of phosphomimetic ProP-PD and bioinformatics, it is possible to suggest links from the kinase (the writer) to the reader (the binding domain) that can be further tested in cell-based assays.

The PDZ domains of DLG1 instead preferentially bind to substrates predicted to be phosphorylated by GRK, PKA, and proto-oncogene serine/threonine protein (Pim3_pim1_group) kinases (corrected *P*-values 0.03), and especially CAM-II substrates (corrected *P*-value 0.0005). In these cases, phosphorylation is expected to disable interactions as the phosphomimetic mutations of the residues confer loss of interactions. Interestingly, the localization of DLG1 itself has been proposed to be regulated by CAMK2D and they have been found in complex (Koh *et al*, 1999).

## On the generality of PDZ domains and phosphopeptide interactions

The literature on ligands containing Ser/Thr phosphorylation sites that positively regulate PDZ domain interactions is sparse. Through our study, we found that Scribble PDZ1 interacts with p-1 and p-3 phosphorylated ligands, and the phosphomimetic ProP-PD results

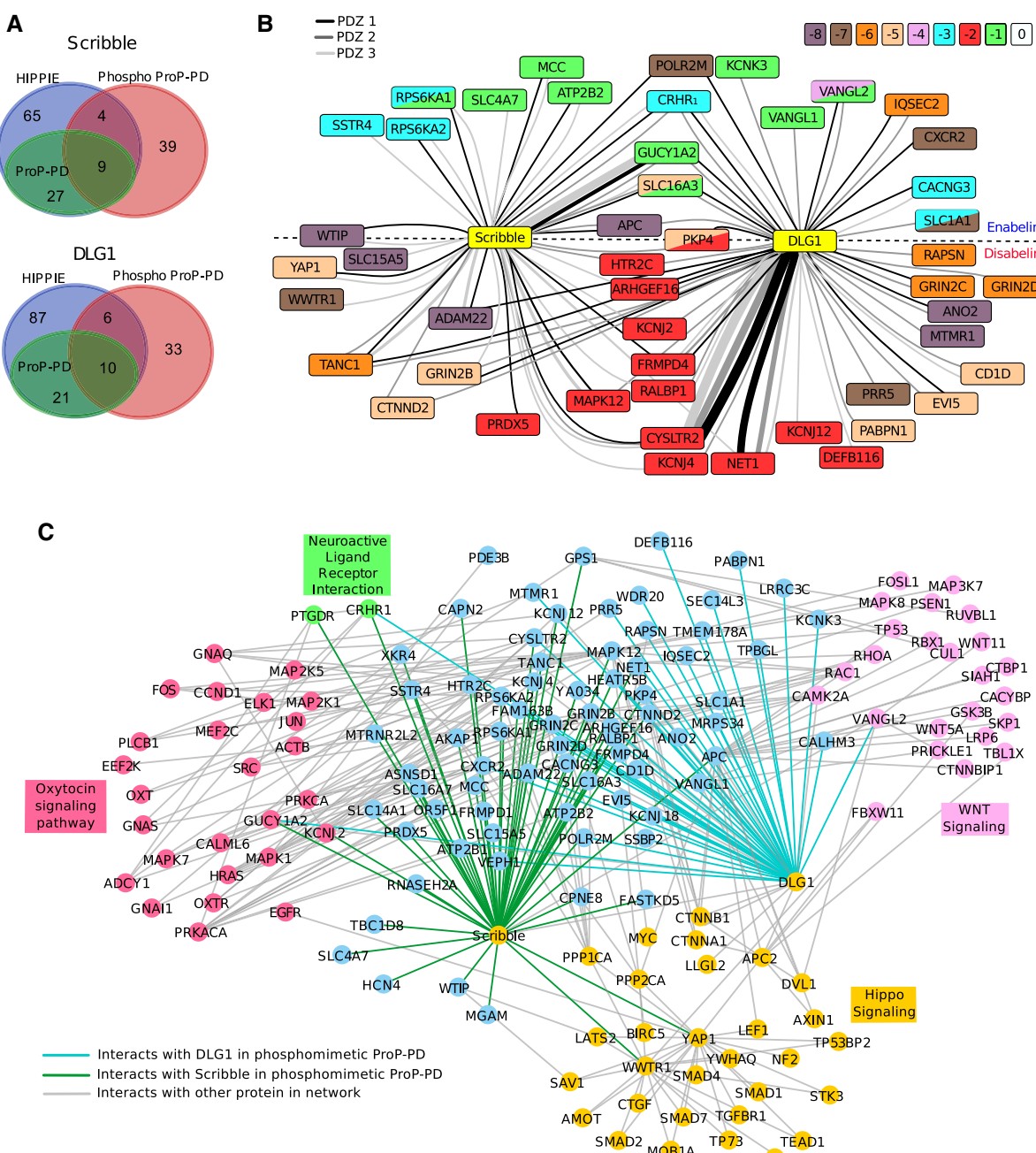

**Figure 4. Bioinformatics analysis of Scribble and DLG1 interactions identified through phosphomimetic ProP-PD.**

A Overlap between ligands identified through phosphomimetic ProP-PD (red) and interactions reported in the HIPPIE database (blue). The HIPPIE database already contains ligands previously reported through ProP-PD (green; Ivarsson *et al*, 2014) for Scribble and DLG1.

B Network representation of high-confidence Scribble and DLG1 ligands. The node color indicates the position of the known or predicted phosphorylation site. Double mutants are indicated by two colors within one node. The edge color indicates the domain (PDZ1, PDZ2, PDZ3) that was found to interact with the target. The edge thickness reflects the total count of the peptide (WT and phosphomimetic) in the NGS analysis. For targets above the dotted line, the phosphomimetic mutations enable interactions. For targets below the dotted line, interactions are disabled by mutations. Interactions with targets on the dotted line are enabled or disabled depending on which PDZ domain they interact with.

C Extended network containing interactions with STRING score over 0.8 to any target protein that are present in KEGG pathways enriched in the STRING-based enrichment analysis.

suggest that Scribble PDZ2 and PDZ3 have similar preferences. We therefore explored the potential phospho-regulation of the interactions through a qualitative analysis of an additional set of nine class

I binding PDZ domains using our phosphomimetic ProP-PD approach (Fig 5A and B). As expected, we found that the interactions of all domains tested are negatively affected by

phosphomimetic mutation of p-2. Phosphomimetic mutation of p-1 was beneficial for six domains tested, and of p-3 for two of the domains (SNTB1 and INADL PDZ6). In contrast, SHANK1 PDZ interactions were negatively influenced by phosphomimetic mutations of p-3 and p-5. To confirm the disabling effect of p-3 phosphorylation on SHANK1 PDZ binding, we determined its affinity for unphosphorylated and phosphorylated RPS6KA2 peptides through MST experiments. The results confirmed the disabling effect of p-3 phosphorylation on SHANK1 binding (3.6×; Fig 5). SHANK1 has previously been shown to interact with RPS6KAs and to be phosphorylated by the kinases (Thomas et al, 2005). Our results suggest that the interaction may be negatively regulated by ligand phosphorylation. RPS6Ks also to interact with other PDZ domains such as Magi PDZ2 that in some cases (e.g., NHERF1) also have been shown to serve as substrates for the kinases (Thomas et al, 2005; Lim & Jou, 2016). In case of Magi PDZ2, it was reported that p-3 phosphorylation had no effect on PDZ binding (Gogl et al, 2018). Phosphorylation of p-3 may thus serve to switch the targets of PDZbms. Although not identified through phosphomimetic ProP-PD, we also determined SHANK1 PDZs' affinity for the p-1 phosphorylated MCC peptide and found that also this interaction is negatively influenced by phosphorylation (Fig 5C, Table EV5). Thus, SHANK1 PDZ interactions appear generally negatively regulated by phosphorylation.

We made a structure-based sequence alignment of the here tested PDZ domains to explore whether the domains that display the same effect on phosphomimetic mutations share common features on the primary sequence level (Fig 5D). In this alignment, we also included the PDZ domain of SNX27, as it was recently reported that ligand phosphorylation (p-3, p-5 or p-6) reinforces its interactions (Clairfeuille et al, 2016). In the alignment, we indicated the residues corresponding to Scribble R762 that we showed is a gatekeeper for Scribble PDZ1's interaction with the p-3 phosphorylated ligand. A basic residue (Arg/Lys) on the corresponding site is present also in SNTB1 and in Scribble PDZ2 and PDZ3 that display a similar pattern of regulation. The presence of a basic residue in this position may thus serve as an indication of putative phosphopeptide binding PDZ domains. Along these lines, a Lys at the corresponding position was previously implicated in the interaction between Tiam1 PDZ and the p-1 Tyr phosphorylated class II binding peptide of syndecan1 (Liu et al, 2013). However, the basic residue is missing in other phosphopeptide PDZ domains such as SNX27 PDZ. SNX27 PDZ has instead an Arg in β2 (corresponding to Scribble A743; indicated in the alignment) that directly lines the peptide binding pocket and is involved in stabilizing the phosphate group of p-3, p-5, or p-6 phosphorylated ligands through a network of residues (Clairfeuille et al, 2016). The SNX27 PDZ domain and Scribble PDZ1 thus accomplish p-3 phosphorylated ligand binding in distinct manners. To explore the plasticity of the phosphopeptide binding, we introduced an Arg in β2 (A743R) in the background of Scribble PDZ1 R762A, which has a preference for unphosphorylated ligands. We determined the affinities for the phosphorylated and unphosphorylated MCC (p-1) and RPS6KA2 (p-3) peptides through ITC (Fig 5E; Table EV5). We found that the R762A/A743R mutation did not confer any specificity for the p-1 phosphorylated MCC peptide, but it restored a preference for p-3 phosphorylated RPS6KA2. Thus, the specificity of Scribble PDZ1 for phosphopeptides can easily be tuned, and it is likely that this plasticity for phosphopeptide binding is shared with several other members of the PDZ domain family. As

the specificity of PDZ domain interactions depends on both direct and indirect interactions (Ernst et al, 2014) and there are multiple ways of the domains to accomplish phosphopeptide binding, it is difficult to draw general conclusions based on the analysis of the primary sequences.

### Pre-phosphorylation ProP-PD

One alternative strategy to identify phosphorylation switches through phage display is to treat the phage library with a kinase prior to phage display selection (Fig 6A). Such a strategy has previously been used to identify ligands of a pTyr binding PTB domain (Dente et al, 1997). To evaluate this approach in comparison with phosphomimetic ProP-PD, we first generated a library that displays the wild-type peptides of the phosphomimetic ProP-PD design (98% of the design was confirmed by NGS). Aiming to test the approach against the bait protein Scribble PDZ1, we then treated this wild-type library with RPS6KA1 and validated the library phosphorylation using a monovalent phospho-Ser-specific antibody (Fig 6B). Finally, we used the phosphorylated and unphosphorylated library in parallel selections against Scribble. Pooled phage ELISA indicated that the selections were successful, and the binding-enriched phage pools were analyzed through NGS (Fig 6C).

Analysis of the enriched peptide sequences revealed that there was no or little difference in terms of the identity of the enriched ligands between the selections. However, there are some significant differences between the two sets. In particular, the TMFLRETSL-coo- peptide of GUCY1A2 obtained significantly larger proportion of the NGS counts in the selection against the pre-phosphorylated library than in selections against the unphosphorylated library (3×). Interestingly, the peptide contains a "R..S" sequence that matches the preferred RPS6KA1 phosphorylation motif (Romeo et al, 2012). Thus, it is plausible that the kinase treatment leads to a phosphorylation of GUCY1A2, which gives a selective advantage consistent with the results of the phosphomimetic ProP-PD. The MAPK12 peptide obtained instead a lower proportion of the NGS counts in the selection against the phosphorylated library, which is likely explained by the fact that it is outcompeted by the phosphorylated GUCY1A2 peptide, as the MAPK12 peptide lacks a suitable RPS6KA1 phosphorylation motif. The results of the phage display in combination with kinase treatment are in good agreement with the results of the phosphomimetic ProP-PD. Indeed, the phosphomimetic TMFLRETEL-coo- peptide dominated the selection against Scribble PDZ1 and occurred three times more than its wild-type counterpart (Fig 2A). Thus, the tandem combination of phosphomimetic ProP-PD and pre-phosphorylation ProP-PD can be a viable approach for identification and validation of phospho-regulated interactions.

## Discussion

In this study, we aimed to test the feasibility of phosphomimetic ProP-PD for the identification of interaction that are regulated by Ser/Thr phosphorylation. We created a phage display library comprising C-terminal peptides in the human proteome with known or putative Ser/Thr phosphorylation sites, and phosphomimetic variants thereof. We demonstrated the power of using such a phosphomimetic ProP-PD library together with NGS to identify

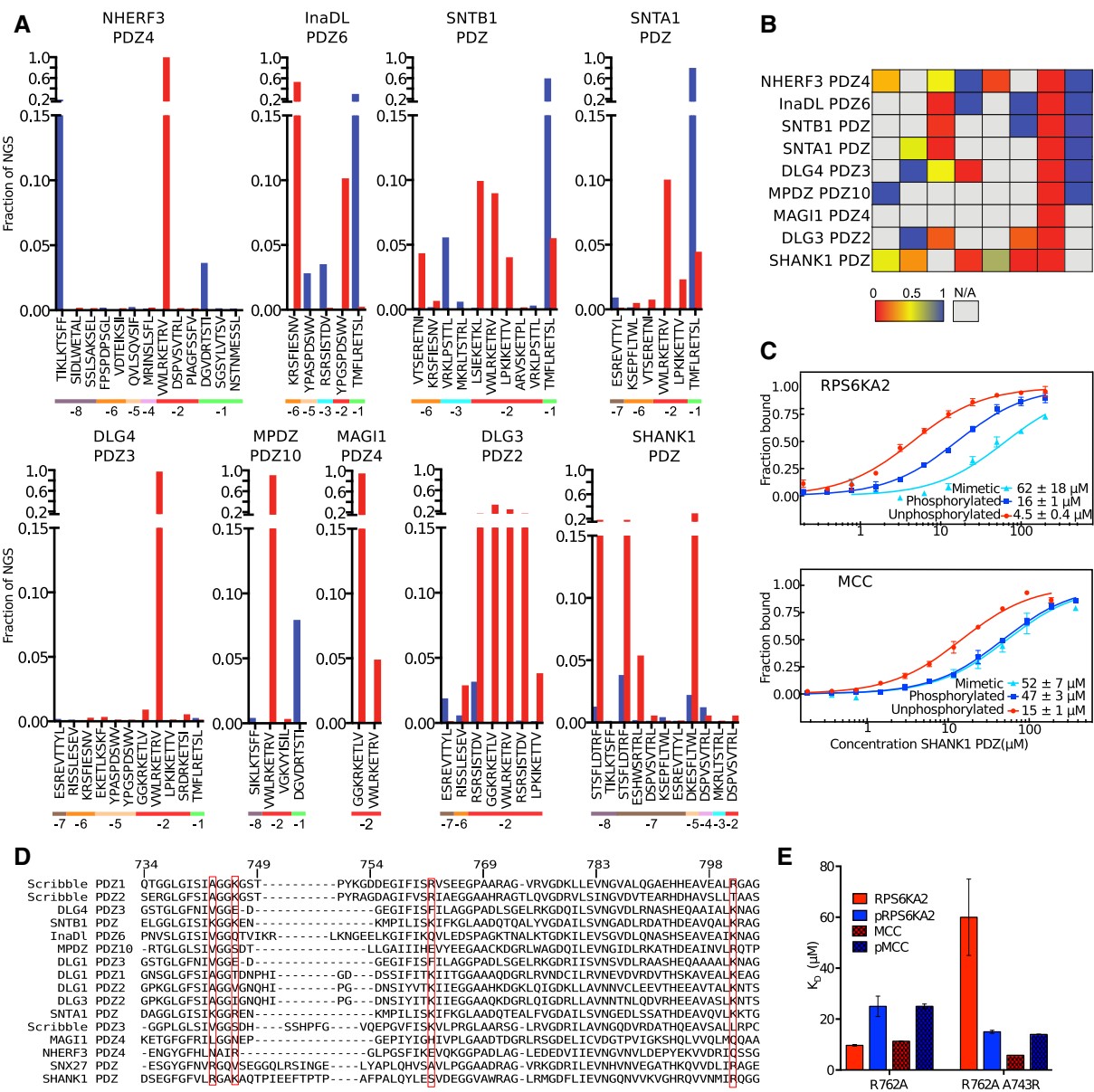

**Figure 5.**  **Qualitative effects of phosphomimetic mutations on interactions with a variety of class I binding PDZ domains as determined by phosphomimetic ProP-PD, together with analysis of the plasticity of the interactions of Scribble PDZ1 with phosphorylated ligands.**

A   Phosphomimetic ProP-PD selection results from selections against nine additional class I binding PDZ domains. The fractions of the sequencing counts from NGS analysis of the binding-enriched phage pools for each peptide wild-type and phosphomimetic peptide pair (wild type in red bars and phosphomimetic peptide in blue). The site of the phosphomimetic mutations is indicated below the peptide sequences. Peptides with multiple phosphorylation sites have been removed for clarity.

B   Scoring matrices of the effects of phosphomimetic mutations on the phosphomimetic ProP-PD results of the PDZ domain scores are calculated from the ratios of NGS counts of the phosphomimetic peptides and the sum of the NGS counts of the wild-type and phosphomimetic peptides. A score of 0 (red) indicates that the selection was dominated by wild-type peptides, and a score of 1 (blue) indicates that the selection was dominated by peptides with phosphomimetic mutations at the given position.

C   MST affinity measurements using FITC-labeled peptides (unphosphorylated, phosphorylated, and phosphomimetic) of MCC (p-1) and RPS6KA2 (p-3) and SHANK1 PDZ. A fixed peptide concentration (25–50 nM) was titrated with varying concentrations of the protein. $K_D$ values were determined using thermophoresis and T-Jump signal for data analysis ($n = 3$; error bars represent SD). Statistical assessment using ordinary one-way ANOVA for multiple comparisons confirmed significant affinity differences between wild-type and phosphomimetic RPS6KA2 ($P \leq 0.01$) and unphosphorylated and phosphomimetic or phosphorylated MCC ($P \leq 0.001$).

D   Structure-based sequence alignment of the PDZ domains explored in this study and SNX27 PDZ that was recently shown to bind phosphopeptides (Clairfeuille *et al*, 2016). The numbering is based on Scribble, and the sequences are ordered based on sequence similarities with Scribble PDZ1. The red boxes indicate the basic residues of Scribble PDZ1 mutated in Fig 3, and residue A743, which aligns with the R58 of the SNX27 PDZ domain that was recently described to be of importance for binding p-3 and p-5 phosphorylated ligands.

E   Histogram of equilibrium dissociation constants of Scribble PDZ1 R762A (mutational background) and the R762A/A743R mutant with unphosphorylated and phosphorylated MCC and RPS6KA2 peptides (error bars represent the SD, $n = 3$). Note that the A743R mutation restores the preference for the p-3 phosphorylated RPS6KA2 peptide.

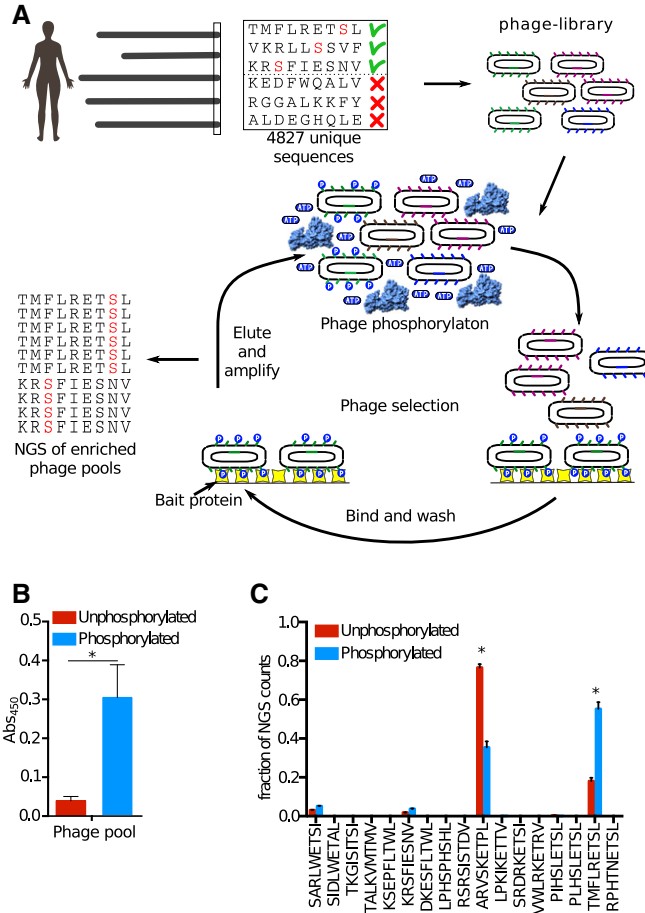

**Figure 6. Pre-phosphorylation ProP-PD confirms phospho-regulation of the interactions of Scribble PDZ1.**

A We constructed a ProP-PD library that displays peptides that represent the C-terminal peptides of the human proteome with known or putative Ser/Thr phosphorylation sites (i.e., the wild-type part of the phosphomimetic ProP-PD library design). The library was subjected to phosphorylation by RPS6KA1 prior to each round of selection against the bait Scribble PDZ1. The binding-enriched phage pools were analyzed by NGS, and the results were compared to the results of a selection against the same library but without phosphorylation.

B Phosphorylation of the ProP-PD library was confirmed through sandwich ELISA. An anti-phospho-Ser/Thr/Tyr antibody was immobilized and challenged with the phosphorylated or unphosphorylated ProP-PD library, and the binding was detected using an HRP-conjugated anti-M13 antibody (error bars represent SD, $n = 2$). A significant difference in the binding to the phospho-antibody was observed for the phosphorylated ProP-PD library as compared to unphosphorylated library ($*P \leq 0.05$, unpaired $t$-test).

C Bar graphs representing the fraction of NGS counts for distinct peptides from the selections against the pre-phosphorylated and unphosphorylated libraries (error bars represent SD, $n = 2$). For peptides indicated with *, there are significant differences between the fractions of NGS counts from the distinct selections, using multiple $t$-tests.

interactions of potential biological relevance that can be enabled and disabled by Ser/Thr phosphorylation. From the perspective of PDZ domains, we provide evidences for a dynamic switch-like mechanism, where phosphorylation at p-1 or p-3 enables interactions with Scribble PDZ domains, while disabling interactions with other PDZ domains, such as SHANK1 PDZ. Such effects are

likely to be of functional importance in a cellular complex where the composition of PDZ protein complexes is determined by the dynamic equilibriums between the relative affinities of the interacting proteins and the relative protein concentrations. Based on our results, we expect that phosphorylation switches of PDZ domain interactions will be more common than previously appreciated.

The rationale behind using Glu as a mimetic mutation of pSer/pThr is that Glu (or Asp) is negatively charged and is nearly isosteric with pSer and pThr (Pearlman et al, 2011). Phosphomimetic mutations have frequently been used to study the effects of phosphorylation since it was shown that replacing a phosphoserine with an Asp mimicked the phosphorylated state of a protein (Thorsness & Koshland, 1987). It also appears as if nature has made use of the reverse engineering by evolving phosphosites from acidic amino acids (Pearlman et al, 2011). It has, however, been less common to use phosphomimetic mutations to study the effects of phosphorylation on a ligand binding site. The success of the approach will depend on the nature of the interactions studied.

On the one side, phosphomimetic mutations serve a qualitative purpose when the intention is to investigate a general charge-dependent interaction although the quantitative effects are not expected to be identical given the differences in net charge at physiological pH (Glu -1, pSer/pThr -2; Cooper et al, 1983) and the differences in hydration shells (Hunter, 2012). Along these lines, we found the effects of phosphomimetic mutations and ligand phosphorylation on PDZ binding to be consistently enabling or disabling depending on the ligand position, although the absolute numbers differed in terms of effects on $K_D$ values. Similar results were reported in a recent study on the effect of p-2 phosphorylation of the EQVSAV-coo- peptide of on binding to the second PDZ domain of PTBL (Toto et al, 2017). In a similar fashion, acidic amino acids or phospho-Ser downstream of the core docking motif of the regulatory subunit B56 of protein phosphatase 2A (L..I.E.) enhances the affinity of the interaction, as also shown by ProP-PD (Hertz et al, 2016; Wu et al, 2017). The phosphomimetic ProP-PD approach should thus be applicable to other cases, for which negative charges contribute to the binding affinity but where there is no strict requirement on phosphorylation. For example, the WD40 repeats form versatile interaction modules that are among the most abundant protein domains. They could represent suitable bait proteins for phosphomimetic ProP-PD as phosphomimetic mutations have been found to enable interactions with WD40 repeat of UAF1 (Villamil et al, 2012) and βTrCP (da Silva Almeida et al, 2012).

On the other side, when it comes to a highly specific phospho-binding site, there is often a strong dependency on the geometry of the Args (or other residues) that line the binding pocket. For example, the FHA domain of Rad53p requires the presence of phospho-Thr and does not tolerate phospho-Ser or Glu (Durocher et al, 1999). For such cases, phosphomimetic mutations are bound to fail. Phosphomimetic mutation has also been found to be less suited for mimicking Ser phosphorylation of the second position of the PP2A B56 binding motif, as its phospho-Ser binding involves specific interactions with an Arg that is too far away to be reached by a phosphomimetic mutation (Wang et al, 2016). Another example for which a phosphomimetic mutation has been shown to fail to mimic the effect of phospho-Ser involves 14-3-3ζ (Zheng et al, 2003).

However, more recently it was found that a phosphomimetic RxEP motif in the NS3 protein of dengue virus interacts with 14-3-3ε (Chan & Gack, 2016). Given the versatility of the binding mode of the 14-3-3 proteins, it is possible that the results will depend on the overall context of the peptide subjected to phosphomimetic mutation. To explore the full potential of phosphomimetic ProP-PD, it will be necessary to design follow-up studies with more complex ProP-PD libraries and to challenge it with a variety of bait proteins.

We further made an attempt to compare phosphomimetic ProP-PD to the pre-phosphorylation ProP-PD where the phage library is phosphorylated with a given kinase prior to selection, which confirmed phospho-regulation of the Scribble PDZ1 binding TMFLRETSL-coo- peptide of GUCY1A2 by RPS6KA1. The approaches could thus be used in tandem. However, the pre-phosphorylation approach would benefit from further optimization. Among the limitations of the approach, as we see it, is that it is unclear what percentage of the phage library is phosphorylated, in particular when using a multivalent display on the major coat protein p8. Furthermore, some peptides contain multiple potential phosphorylation sites, and it might become difficult to distinguish exactly what phosphorylation event is responsible for a shift in the enriched peptides. A more sophisticated approach to tackle PTM-dependent interactions would be to use an expanded genetic code, where an amber stop codon is used to encode a twenty-first additional amino acid. Given that an expanded genetic code has been used for protein evolution experiments with several unnatural amino acids (Liu *et al*, 2008) and that optimized orthogonal aminoacyl-tRNA synthetase/tRNA$_{CUA}$ pairs for phospho-Ser and phospho-Thr incorporation were recently generated (Rogerson *et al*, 2015; Zhang *et al*, 2017), such an approach should be feasible and is likely to become a viable approach in specialized laboratories. For such a setup, it will be crucial to ensure that the full-length phospho-Ser-containing peptides are produced and displayed at the same level as other peptides and that the phospho-Ser displaying phage do not suffer any growth disadvantage. On a related note, in a recent study they encoded human phospho-Ser-containing peptides in bacteria for profiling of phospho-dependent interactions (Barber *et al*, 2018). Thus, this and other emerging approaches will be of complementary use for finding phosphorylation-dependent interactions. Among the methods, phosphomimetic ProP-PD offers an efficient approach for detecting potential phosphorylation switches without the requirement of the usage of alternative codons or optimization of protocols and thus offers a straightforward and attractive approach of broad applicability for exploring the potential phospho-switching of motif-based interactions.

The focus of this study is on the C-terminal regions of the proteome, and interactions with PDZ domains, but it can be envisioned to create libraries that cover a large proportion of the Ser/Thr phosphosites in the disordered regions of the human proteome. Further studies with more extensive phosphomimetic ProP-PD libraries and alternative sets of phosphopeptide binding domains will establish the general applicability of the approach. We expect that the approach will prove to be feasible for probing phospho-regulated interactions on a large scale, in particular for domains that have an inherent affinity for their unphosphorylated ligands that is enhanced by the addition of the extra charges. The approach also offers an efficient way to pinpoint interactions that are negatively regulated by ligand phosphorylation. As discussed, the method is less suited for proteins with strict requirement on the phosphate group for binding. In an expanded form, phosphomimetic ProP-PD can be a highly useful tool for assigning functional roles to the phosphoproteome, and for linking of phosphorylation switched SLiMs to their respective binding domains, thereby contributing to solving the phosphorylation code.

# Materials and Methods

## Reagents and tools table

| Reagent/Resource | Reference or Source | Identifier or catalog number |
|---|---|---|
| **Experimental models** | | |
| HEK293 Human embryonic kidney cells | Sigma Aldrich | 85120602 |
| *E. coli* SS320 | Lucigen | 60512-1 |
| *E. coli* omniMAX | Fisher Scientific | C85812-01 |
| *E. coli* BL21(DE3) | Sigma Aldrich | CMC0014 |
| **Recombinant DNA** | | |
| pDONR223-RPS6KA2 | Addgene (Johannessen *et al*, 2010) | # 23530 |
| pKH3-human RSK1 | Addgene (Richards *et al*, 2001) | # 13841 |
| p3XFLAG-CMV^a-10 Expression Vector | Sigma Aldrich | E7658 |
| GFP-tagged full-length human Scribble | JP Borg (INSERM Marseille) | N/A |
| Scribble PDZ1 petM11 PDZ1 aa 719-829 Human | Ivarsson *et al* (2014) | N/A |
| Scribble PDZ1 pGEX Human | Tonikian *et al* (2008) | N/A |
| Scribble PDZ2 pGEX Human | Tonikian *et al* (2008) | N/A |
| Scribble PDZ3 pGEX Human | Tonikian *et al* (2008) | N/A |

**Reagents and Tools table** (Continued)

| Reagent/Resource | Reference or Source | Identifier or catalog number |
|---|---|---|
| DLG1 PDZ1 pGEX Human | Tonikian *et al* (2008) | N/A |
| DLG1 PDZ2 pGEX Human | Tonikian *et al* (2008) | N/A |
| DLG1 PDZ3 pGEX Human | Tonikian *et al* (2008) | N/A |
| NHREF3:4 pGEX Human | Tonikian *et al* (2008) | N/A |
| Magi1 PDZ4 pGEX Human | Tonikian *et al* (2008) | N/A |
| DLG3 PDZ2 pGEX Human | Tonikian *et al* (2008) | N/A |
| DLG4 PDZ3 pGEX Human | Tonikian *et al* (2008) | N/A |
| Inadl PDZ6 pGEX Human | Tonikian *et al* (2008) | N/A |
| MPDZ PDZ10 pGEX Human | Tonikian *et al* (2008) | N/A |
| SHANK1 PDZ pGEX Human | Tonikian *et al* (2008) | N/A |
| SNTA1 PDZ pGEX Human | Tonikian *et al* (2008) | N/A |
| SNTB1 PDZ pGEX Human | Tonikian *et al* (2008) | N/A |
| DLG1 PDZ 2 311–407 in a modified pRSET vector 40 Human | Chi *et al* (2009) | N/A |
| M13 major coat protein P8 phagemid | Held and Sidhu (2004) | N/A |
| **Antibodies** | | |
| *Rabbit anti-FLAG* | Abcam | Cat # ab 1162 |
| *Rabbit anti-GFP* | Abcam | Cat # ab 6556 |
| Rabbit anti-beta Actin | Abcam | Cat # ab 1801 |
| HRP-conjugated anti rabbit secondary antibody | Life Technologies | ref # G21234 |
| HRP-ANTI-M13 monoclonal conjugate | GE Healthcare | 27-9421-01 |
| Mouse Anti Phospho Ser/Thr/Tyr antibody | Invitrogen | MA1-38450 |
| **Oligonucleotides and sequence-based reagents** | | |
| *Library primers* | This study | Table EV2 |
| AATGATACGGCGACCACCGAGATCTACACTCTTTCCCTACACGACGCTCTTCCGATCT(8 NT barcode)TATGCAGCCTCTTCATCTGGC | This study | N/A |
| CAAGCAGAAGACGGCATACGAGCTCTTCCGATCT(6 NT barcode)CCTCCGGATCCTCCACC | This study | N/A |
| CGAAGGCATCTTCATCTCTGCTGTGTCTGAAGAAGGTCC | This study | R7762A Scribble PDZ1 F |
| GGACCTTCTTCAGACACAGCAGAGATGAAGATGCCTTCG | This study | R7762A Scribble PDZ1 R |
| CATCGCCGGCGGTGCAGGCTCCACTCC | This study | K746A Scribble PDZ1 F |
| GGAGTGGAGCCTGCACCGCCGGCGATG | This study | K746A Scribble PDZ1 R |
| GTCTGGGTATCAGCATCAGAGGCGGTAAAGGCTCCAC | This study | A743R Scribble PDZ1 F |
| GTGGAGCCTTTACCGCCTCTGATGCTGATACCCAGAC | This study | A743R Scribble PDZ1 R |
| CAGTAGAAGCGCTGGCTGGTGCGGGCACTG | This study | R801A Scribble PDZ1 F |
| CAGTGCCCGCACCAGCCAGCGCTTCTACTG | This study | R801A Scribble PDZ1 R |
| GTGAAGCTTATGGACCTGAGCATGAAGAAGTTCGC | This study | KS6A2 cloning primer F |
| CTAGAATTCACAACCGCGTGGACGTGAGTCTCTTC | This study | KS6A2 cloning primer R |
| CTAGAATTCACAACCGCGTCTCCGTGAGTCTCTTC | This study | KS6A2RS>E cloning primer R |
| CTAGAATTCACAACCGCGTGGCCGTGAGTCTCTTC | This study | KS6A2RS>A cloning primer R |
| TCCGAATTCATGCCGCTCGCCCAGCTCAAGGAGCC | This study | KS6A1 cloning primer F |
| CCTGATATCACAGGGTGGTGGATGGCAACTTCCTC | This study | KS6A1 cloning primer R |
| CCTGATATCGGCCATCGATCAGGATGGCAACTTC | This study | KS6A1-TTL cloning primer R |
| CTAGAATTCGAAAGTTGGGTATCAGGACGTGAGTCTC | This study | KS6A2-TRL cloning primer R |
| CCTGATATCACAGGGTGGTCTCTGGCAACTTCCTC | This study | KS6A1R1E cloning primer R |
| CCTGATATCACAGGGTGGTGGCTGGCAACTTCCTC | This study | KS6A1R1A cloning primer R |

**Reagents and Tools table** (Continued)

| Reagent/Resource | Reference or Source | Identifier or catalog number |
|---|---|---|
| **Chemicals, enzymes and other reagents** | | |
| T4 PNK | Fisher Scientific | 10531061 |
| T7 pol, 300 units | Fisher Scientific | 10529710 |
| T4 DNA ligase | Fisher Scientific | 10723941 |
| Exonuclease I | VWR | E70073Z |
| Hypure molecular biology grade water | GE healthcare | SH30538.03 |
| Shrimp alkaline phosphatase, 1,000 units | VWR | E70092Z |
| Phusion | Fisher Scientific | 10402678 |
| DNASE I 1 × 100 mg | vwr | A3778.0100 |
| Glutathione Magnetic Beads 20 ml | Fisher Scientific | 11834131 |
| Anti-FLAG® M2 Magnetic Beads | Sigma Aldrich | M8823 |
| FITC-HTNETEL | GeneCust | N/A |
| FITC-HTNETSL | GeneCust | N/A |
| FITC-HTNETpSL | GeneCust | N/A |
| RPHTNETSL | GeneCust | N/A |
| RPHTNETpSL | GeneCust | N/A |
| MKRLTSTRL | GeneCust | N/A |
| MKRLTpSTRL | GeneCust | N/A |
| FITC-RLTSTRL | GeneCust | N/A |
| FITC-RLTETRL | GeneCust | N/A |
| FITC-RLTpETRL | GeneCust | N/A |
| FITC-KRSFIESNV | GeneCust | N/A |
| FITC-KREFIESNV | GeneCust | N/A |
| FITC-KRpSFIESNV | GeneCust | N/A |
| ARVSKETPL | GeneCust | N/A |
| ARVSKEpTPL | GeneCust | N/A |
| Complete™, EDTA-free Protease Inhibitor Cocktail | Sigma Aldrich | 4693132001 |
| DMEM, high glucose, GlutaMAX™ Supplement | Fisher Scientific | 61965026 |
| Fetal Bovine Serum, qualified, US origin | Fisher Scientific | 26140087 |
| MEM Non-Essential Amino Acids Solution (100×) | Fisher Scientific | 11140035 |
| Trypsin-EDTA (0.5%) | Fisher Scientific | 15400054 |
| Fugene HD | Promega | E2311 |
| HindIII | Fisher Scientific | ER0501 |
| RPS6KA1 | Thermo Fisher | PV3680 |
| EcoRI | Fisher Scientific | ER0271 |
| AcTEV | Fisher Scientific | 10216572 |
| SmaI fast digest | Fisher Scientific | 10324630 |
| Taq DNA Polymerase | VWR | 733-1819 |
| **Software** | | |
| Cytoscape v3.6.1 | http://www.cytoscape.org Shannon *et al* (2003) | |
| R 3.4.2 | R Core Team (2017) https://www. R-project.org/ | |
| STRING 10.2 | Szklarczyk et al Nucleic Acids Res. 2015 43(Database issue):D447-52 | |

**Reagents and Tools table** (Continued)

| Reagent/Resource | Reference or Source | Identifier or catalog number |
|---|---|---|
| Networkin 3.0 | Horn et al, KinomeXplorer: an integrated platform for kinome biology studies. Nature Methods 2014;11(6):603–604 | |
| Perl v5.18.2 | http://www.perl.org/ | |
| PRISM 6 | https://www.graphpad.com/support/prism-6-updates/ | |
| Netphos 3.2 | Blom et al (2004) | |
| Origin 7 | OriginLab corporation | |
| MO. Affinity Analysis v2.1.2 | NanoTemper | |
| Venn Diagram | http://bioinformatics.psb.ugent.be/webtools/Venn/ | |
| **Other** | | |
| Monolith NT automated MST | NanoTemper | |
| ITC200 | Malvern Instruments | |
| 1× MiSeq sequencer (Illumina) | Illumina | |
| NMR 600, and 900 MHz | Bruker | |

## Library design

The phosphomimetic ProP-PD library was designed to contain C-terminal peptides of the human proteome with known and predicted phospho-Ser/Thr sites.

1   Select human C-terminal (9mers) sequences from human SwissProt/UniProt containing known Ser or Thr phosphosites from PhosphoSite (Hornbeck *et al*, 2004, 2015), Phospho-ELM (Dinkel *et al*, 2011), and NetPhos 3.1 (Blom *et al*, 2004). For this study, the data were downloaded on October 10, 2013.

2   Design a phage library that contains all the selected wild-type sequences and phosphomimetic variants thereof (Ser/Thr to Glu). Add combinatorial entries in case of multiple predicted or measured sites: If, for example, two phosphorylation sites (A and B) are found in the same sequence, add pA + Bw, Aw + pB, and pA + pB (where "p" denotes the phosphomimetic variant and w the wild type). The design for this study contains a total of 12455 entries (Table EV1). In case of the Phospho-ELM and NetPhos predictions, kinase information is provided in the library design file when available.

3   Generate the oligonucleotide sequences required for library construction by reversely translating the peptide sequences using codons optimized for *E. coli* expression. Add flanking annealing sites complementary to the phagemid vector (Table EV2).

## Library construction

The M13 phage library was constructed based on a phagemid coding for M13 major coat protein p8 modified for C-terminal peptide display (Held & Sidhu, 2004), and with a *SmaI* restriction site added between the annealing sites used for library construction.

1   Produce single-stranded DNA as described in Rajan and Sidhu (2012).

2   Amplify the obtained oligonucleotide library (CustomArray) by PCR using Phusion DNA polymerase according to the protocol (Thermo Scientific). Remove residual single-stranded DNA (ssDNA) by ExoI treatment (0.2 units/μl, 37°C for 30 min, 85°C for 15 min) followed by flash-cooling on ice. Purify the PCR product with QIAquick Nucleotide Removal Kit (Qiagen). Quantify the PCR product using PicoGreen dsDNA quantitation assay (Thermo Fisher).

3   Phosphorylate 0.6 μg of the PCR product with 20 units of T4 polynucleotide kinase for 1 h in TM buffer (10 mM $MgCl_2$, 50 mM Tris–HCl, pH 7.5) supplemented with 1 mM ATP and 5 mM DTT.

4   Directly after phosphorylation, anneal the primers to 20 μg of the ssDNA phagemid (90°C for 3 min, annealing at 50°C for 3 min and 20°C for 5 min) in TM buffer in a total volume of 250 μl.

5   Synthesize dsDNA by the addition of 10 μl 10 mM ATP, 10 μl 10 mM dNTP, 15 μl 100 mM DTT, 30 Weiss units of T4 DNA ligase, and 30 units of T7 DNA, at 20°C for 16 h. Stop the reaction by freezing the reaction mixture (−20°C).

6   Thaw the sample and digest wild-type DNA using SmaI (Thermo Fisher) for 3 h at 30°C.

7   Purify the dsDNA by QIAquick DNA purification kit (Qiagen). Elute it with 35 μl ultrapure $H_2O$ (HyPure; GE Healthcare).

8   Electroporate the purified dsDNA into *E. coli* SS320 (Lucigen) preinfected with M13KO7 helper phage (New England Biolabs; for preparation of the bacteria, see Rajan & Sidhu, 2012) using a Gene Pulser (Bio-Rad) electroporation system at an electric potential of 2.5 kV, a capacitance of 25 μFD, and a resistance of 100 Ω.

9   Immediately add the bacteria to 25 ml preheated SOC media (0.5 W/V% yeast extract, 2 W/V % tryptone, 10 mM NaCl, 2.5 mM KCl, 10 mM $MgCl_2$, 10 mM $MgSO_4$, and 20 mM glucose), and incubate for 30 min.

10  Determine the electroporation efficiency by titering the phagemid containing bacteria through serial dilutions. Sample 10 µl of the bacterial culture and make 10× serial dilutions in 2YT (0.5 W/V% NaCl, 1 W/V% yeast extract, and 1.6 W/V% tryptone). Spot 5 µl/dilution of each dilution on LB agar plates supplemented with 100 µg/ml carbenicillin. Incubate the plates overnight at 37°C. Count the number of colonies. For this study, the transformation efficiencies were $5 \times 10^6$ transformants for the phosphomimetic ProP-PD library and $1 \times 10^6$ transformants for the wild-type library, which more than 100-fold exceeds the number of unique sequences in the library design.

11  Add the bacterial culture (from point 10) to 500 ml 2YT and incubate overnight at 37°C with shaking.

12  Pellet the bacteria (15 min, 3,500 $g$) and transfer the phage supernatant to new centrifugal tubes containing one-fifth the final volume of PEG/NaCl (20% PEG 8000, 400 mM NaCl) for precipitation of the phage. Incubate for 5 min on ice and pellet the phage by centrifugation (25,000 $g$ for 20 min) at 4°C.

13  Resuspend the phage pellet in 10 ml phosphate-buffered saline (PBS; 37 mM NaCl, 2.7 mM KCl, 8 mM $Na_2HPO_4$, and 2 mM $KH_2PO_4$), pH 7.4, with 0.05% Tween-20.

14  Remove insoluble debris by centrifugation for 10 min at 25,000 $g$. Transfer the phage solution to a clean Falcon tube.

15  Supplement the library with 20% (vol/vol) glycerol, and store it at −80°C.

16  Estimate the phage concentration by infecting actively growing *E. coli* OmniMAX ($OD_{600}$ = 0.6) with serial dilution of the library. Spot 5 µl of each dilution on LB agar plates supplemented with 100 µg/ml carbenicillin, and incubate overnight. The phage concentration of the phosphomimetic library was estimated to be $10^{10}$ colony-forming units (cfu)/µl, and the concentration of the wild-type library was estimated to be $1.8 \times 10^9$ cfu/µl.

### Protein expression and purification

Glutathione transferase (GST)-tagged PDZ domains and GST for phage display selections were expressed and purified as described previously (Tonikian *et al*, 2008; Ivarsson *et al*, 2014). Protein expression for MST, ITC, and NMR experiments was as follows: Plasmids encoding His-tagged proteins (human Scribble PDZ1 aa 719–829 in pETM-11, human DLG1 PDZ2 aa 311–407 in a modified pRSET vector 40, or human Shank1 PDZ aa 652–761 in pETM-33) were transformed into *E. coli* BL21 Gold (DE3; Agilent), and the proteins were expressed in 2YT media supplemented with appropriate antibiotic (70 or 100 µg/ml carbenicillin). Protein expression was induced upon reaching an $OD_{600}$ of 0.8 by adding 1 mM isopropyl β-D-1-thiogalactopyranoside (IPTG) final concentration and allowed for 4 h at 30°C.

Scribble PDZ1 for NMR experiments was expressed in M9 salt enriched with 13C D-glucose and 15N ammonium chloride. Bacteria were harvested by centrifugation at 10,000 × $g$ for 20 min at 4°C. The pellets were resuspended in PBS supplemented with 5 mM $CaCl_2$, 10 µg/ml DNaseI, and 1 mg/ml lysozyme and protease inhibitor (cOmplete Mini EDTA-free; Sigma-Aldrich) and incubated for 1 h at 4°C. The lysates were subjected to sonication in 1- to 3-s pulses for 20 pulses. The lysate was cleared by centrifugation at 20,000 × $g$ for 1 h. The cleared lysates were supplemented with 20 mM imidazole and incubated 1 h with $Ni^{2+}$ Sepharose resin (GE Healthcare). The resin was washed with PBS and 50 mM imidazole, pH 7.4. Bound proteins were eluted in PBS with 300 mM imidazole and dialyzed to appropriate experimental buffers using SnakeSkin dialysis membrane (50 mM Tris–HCl, pH 7.4, for MST; and 20 mM HEPES, pH 7.4, for ITC).

For MST affinity measurements involving SHANK1 PDZ, the His-GST-tag was removed using GST-tagged HRV3C protease. The protein was mixed with 1:100 (w/w) HRV3C protease and dialyzed overnight into PBS (pH 7.4) containing 10 mM $MgCl_2$, 0.05% Tween-20, and 1 mM B-mercaptoethanol. The following day, the cleaved His-GST tag and the GST-HRV3C protease were removed using Glutathione Sepharose 4B resin (GE Healthcare), and the supernatant containing untagged SHANK1 PDZ was collected.

For NMR experiments, we cleaved the His-tag of the labeled protein overnight at 8°C with 0.1 µg of His-tagged TEV protease per mg Scribble PDZ1, while dialyzing in 50 mM Tris–HCl (pH 8.0), 0.5 mM EDTA, and 1 mM DTT. The untagged Scribble PDZ1 was collected as a flow in an $Ni^{2+}$ Sepharose column and was dialyzed into 10 mM $NaPO_4$, pH 6.8. The sample was flash-frozen and stored at −20°C until further use.

### Phosphorylation of wild-type ProP-PD library

For pre-phosphorylation of the wild-type ProP-PD library, we obtained activated RPS6KA1 (Thermo Fisher). The time and amount of kinase required to phosphorylate the ProP-PD library were tested in a phosphorylation assay.

1   Immobilize 1 µg/well anti-phospho-Ser/Thr/Tyr antibody (MA1-38450; Invitrogen) or 0.5% BSA in Kinase buffer (50 mM HEPES (pH 7.5), 10 mM $MgCl_2$, and 1 mM EGTA) at a final volume of 100 µl in a 96-well MaxiSorp microtiter plate (NUNC) overnight at 4°C. Dilute 10 µl of phage library per trial in PBS and precipitate it by adding one-fifth the final volume of PEG/NaCl, incubate on ice for 10 min, and pellet the phage by centrifugation at 12,000 $g$ for 20 min.

2   Resuspend the phage pellet in Kinase buffer and add ATP (100 µM final concentration) and kinase (200 or 400 ng/ml, final concentration) to the phage solution. As a negative control, perform identical reactions without addition of kinase alongside the phosphorylation reactions.

3   Incubate the reactions in room temperature and stop the reactions after 1 or 2 h by heat inactivation of the kinase for 10 min at 60°C.

4   Remove the liquid from the microtiter plate and block the wells by adding 200 µl kinase buffer with 0.5% BSA. Incubate for 2 h at 4°C.

5   Wash the wells with the immobilized antibody or BSA four times with KT buffer (Kinase buffer + 0.05% Tween-20).

6   Add the phosphorylated library or the negative control (the unphosphorylated library) to the wells with immobilized antibody and to the wells with immobilized BSA. Incubate for 2 h at 4°C.

7   Wash the plates four times with KT buffer.

8   To detect bound phage, add 100 µl anti-M13 antibody (GE Healthcare) diluted 1:5,000 in Kinase buffer supplemented with 0.05% Tween-20 and 0.5% BSA and incubate for 1 h at 4°C.

9   Wash the plate four times with KT buffer and one time with Kinase buffer.

                                                                   

10  Add 100 μl of TMB substrate per well, and allow the reaction to develop for 5 min and then stop it by adding 100 μl 0.6 M $H_2SO_4$.

11  Detect the absorbance at 450 nM using a SpectraMax Plus (Molecular Devices) plate reader.

## Phage display selections

Phage selections were carried out for four rounds following a protocol by Huang and Sidhu (Huang & Sidhu, 2011) with minor modifications. Non-specific binding phage was removed by pre-selections against immobilized GST throughout the selection. The selections against the (pre-)phosphorylated wild-type library were performed in 50 mM HEPES (pH 7.4) and 150 mM NaCl (HBS) following the same procedure after pre-phosphorylation of the library.

1  Immobilize proteins (15 μg in 100 μl PBS) in a 96-well Maxi-Sorp microtiter plate (NUNC) overnight at 4°C.

2  Start a 2YT culture of *E. coli* Omnimax supplemented with 100 μg/ml tetracycline to retain the F pilus epitope.

3  Block the wells with 200 μl 0.5% BSA in PBS for 2 h at 4°C.

4  Precipitate naïve phage library ($10^{11}$ cfu per selection) or amplified phage pool from the previous round of selection by adding one-fifth the volume of PEG/NaCl.

5  Incubate on ice for 10 min, and pellet precipitated phage by centrifugation at 5,300 *g* for 15 min. Resuspend the phage pellet in 100 μl PBS per selection.

6  Add 100 μl of phage solution to the GST-coated pre-selection wells and incubate for 1 h.

7  Transfer the phage solution to the immobilized target proteins and allow it to bind for 2 h at 8°C.

8  Wash the plate four times with 200 μl cold wash buffer (PBS, 0.05% Tween-20).

9  Elute bound phage by incubation with 100 μl log phase (0.6–0.8 $OD_{600}$) *E. coli* Omnimax for 30 min at 37°C with shaking (200 RPM).

10  Add $10^{12}$ cfu/ml M13K07 helper phage for hyperinfection of the bacteria and incubate as above for 45 min.

11  Transfer the bacterial culture to 10 ml 2YT supplemented with 50 μg/ml carbenicillin, 25 μg/ml kanamycin, and 300 μM IPTG and incubate overnight at 37°C while shaking.

12  Perform phage pool ELISA to monitor the progress of the selection as described previously (Huang & Sidhu, 2011).

## Pre-phosphorylation of phage library

1  Precipitate the naïve wild-type phage library (as described above, $1.8 \times 10^{10}$ cfu phage per selection).

2  Resuspend the precipitated phage pellets in 100 μl Kinase buffer per selection and add ATP (100 μM final concentration) and RPS6KA1 (500 ng/ml) to the phage solution. In parallel, prepare a negative control where only ATP is added to the phage pool.

3  Incubate the phosphorylation reaction for 2 h at room temperature. Stop the reaction through PEG/NaCl precipitation (as described above).

4  Resuspend the phage pool in 100 μl HBS.

During the 4 days of pre-phosphorylation ProP-PD selections, the input phage library of each day of selection was subjected to phosphorylation prior to phage selection.

## Next-generation sequencing and analysis of phage selection

The naïve library and the enriched phage pools of the third day of selection were amplified and barcoded by PCR and analyzed by NGS as described elsewhere (McLaughlin & Sidhu, 2013; Ivarsson *et al*, 2014). The reads were demultiplexed using the barcode sequences and filtered by discarding sequences with a Phred quality score under 30. Sequences were mapped to the original library using Bowtie (Langmead *et al*, 2009) with two mismatches between reads and library sequence. Nucleotide mutations leading to amino acid substitutions were subsequently removed. Cutoff values of NGS counts were assigned for each pool. PWMs were generated based on the remaining peptides (aligned on the C-termini) using WebLogo (Crooks *et al*, 2004).

A correlation (Pearson correlation as implemented in R) analysis of the sequences obtained from the binding-enriched phage pools with the sequencing counts of the initial library showed that there is a higher probability that a peptide that is present in high count in the naïve library to be selected during the phage display at all and therefore a relationship between the representation of a peptide in the library to the screens sensitivity (Fig EV1). However, there was no significant correlation of the actual read counts (selection "strength") and the initial representation in the library, reflecting the strong selection during phage display (wt+mimetic: *P*-value: 0.49, ρ = −0.08, mimetic only: *P*-value: 0.069, ρ = −0.2, wt only: *P*-value: 0.12, ρ = 0.17). Consequently, the amount of reads for each binder after phage display is not a function of the amount of input reads in the naïve library and no modeling based on the input library frequencies to score the interactions is necessary. To allow for comparison between replicate selections, we used the fraction of NGS reads within a certain set, instead of the absolute number of NGS reads, as the absolute numbers differ between different NGS batches.

## Cloning of constructs and mutagenesis

pDONR223-RPS6KA2 (Johannessen *et al*, 2010) was a gift from William Hahn & David Root (Addgene plasmid # 23530). The gene was PCR-amplified introducing *HindIII* and *EcoRI* restriction sites along with the C-terminal phosphomimetic (S730E) or phosphoinhibiting (S730A) mutations, or the PDZbm deletion (T731-L733). pKH3-human RSK1 was a gift from John Blenis (Addgene plasmid # 13841; Richards *et al*, 2001). The gene was PCR-amplified introducing *HindIII* and *EcoRI* restriction sites, and a and PDZbm deletion (T733-L735) was generated. The genes (wild type and mutants) were cloned into 3xFlag CMV-10 vector and subjected to Sanger sequencing. Site-directed mutagenesis to introduce the single-point mutations K746A, K762A, R762A, and R801A in Scribble PDZ1 in pETM-11 was performed using the QuikChange protocol.

## MST affinity measurements

Affinities for wild-type, phosphorylated, and phosphomimetic FITC-labeled synthetic peptides (TANC-1: KR(S/pS/E)FIESNV; RPS6KA2: RLT(S/PS/E)TRL; MCC: HTNET(S/PS/E)L) were obtained from GeneCust, and the peptides were dissolved in 100% dimethyl sulfoxide (DMSO) to a concentration of 1–5 mM. The peptides were diluted in 50 mM Tris–HCl, pH 7.4, and 0.1% Tween-20 for affinity measurements with Scribble PDZ1 and DLG PDZ2. For Shank1 PDZ,

the peptides were diluted in PBS, 10 mM MgCl$_2$, and 0.05% Tween-20. The protein was subjected to a 1:1 dilution series in a 384-well microtiter plate. Peptides were added to a final concentration in the range of 25–50 nM. The samples were run in standard capillaries with a blue filter on a Monolith NT.Automated MST (NanoTemper) with MST power 20 or 40. The experiments were carried out in triplicate using the same batch of protein and peptides. Statistical assessment was made by one-way ANOVA comparing the change in affinity for the mimetic and phosphorylated peptide to the unphosphorylated peptide.

## ITC experiments

Peptides and proteins were dialyzed in the same beaker in 20 mM HEPES, pH 7.4. The experiments were carried out in 25°C in an iTC200 (Malvern Instruments) with a protein concentration in the cell of 50–100 μM and between 0.64 and 3 mM peptide in the syringe. All experiments were done in triplicate with the same batch of protein and peptides. The results were analyzed using Origin 7 (OriginLab Corporation). The peptide to protein stoichiometry was fixed between replicates to 0.8–1.1 (depending on peptide) and the baseline was corrected to get a low chi value in the curve fit, using single-site binding. Statistical assessment of binding affinity between unphosphorylated and phosphorylated peptides was done by unpaired Student's *t*-test comparing the mean ± SD between the triplicate titrations.

## NMR experiments

NMR experiments were recorded on Bruker 600- and 900-MHz spectrometers, equipped with triple-resonance cryogenic temperature probes at 298 K. The protein complexes were formed by titrating unlabeled phosphopeptide (pRPS6KA2) until saturating amounts with labeled Scribble PDZ1 domain (500 μM). Prior to the titration, 0.01% NaN$_3$ and 5% D$_2$O were added to the samples. For assignment of the protein backbone, 2D [$^1$H,$^{15}$N]-HSQC, 3D HNCACB, and [$^1$H,$^1$H]-NOESY-[$^1$H,$^{15}$N]-HSQC experiments (Cavanagh *et al*, 2007) were performed. Side chains were assigned using the following 3D experiments: [$^1$H,$^{15}$N]-HSQC-[$^1$H$^{13}$C-$^{13}$C$^1$H]-TOCSY (60 ms mixing time), [$^1$H,$^1$H]-NOESY-[$^1$H,$^{13}$C]-HSQC, [$^1$H,$^{13}$C]-HSQC-[$^1$H$^{13}$C-$^{13}$C$^1$H]-TOCSY (60 ms mixing time) in combination with 2D [$^1$H,$^{15}$N]-HSQC, and 2D [$^1$H,$^{13}$C]-HSQC-CT(REFS). ϕ-angle-restraining $^3J_{HNHA}$ couplings were determined from a 3D HNHA-type experiment using quantitative *J*-coupling intensity evolution (Vuister & Bax, 1994). Distance restraints were collected in the following NOESY experiments, and the following parameters were used: 3D NOESY-[$^1$H,$^{15}$N]-HMQC and NOESY-[$^1$H,$^{13}$C]-HMQC with 60 ms mixing time and 128 ($^{15}$N or $^{13}$C) × 200 ($^1$H) × 2,048 ($^1$H, direct) number of points (Cavanagh *et al*, 2007). Standard 2D HSQC titration experiments were made for phosphorylated and unphosphorylated MCC and MAPK12 peptides as above.

## Co-immunoprecipitations and GST-pulldowns

The plasmids encoding 3xFlag RPS6KA2 wild type and mutants thereof, 3xFlag RPS6KA1 and mutants thereof, and GFP-tagged full-length human Scribble (Navarro *et al*, 2005) were Midiprepped using PureLink HiPure Plasmid Midiprep Kit (Invitrogen). HEK293 cells were maintained in DMEM supplied with 10% fetal bovine serum and 1× non-essential amino acids and 1% Pen/Strep (Gibco). Prior to transfection, the cells were plated with a density of $5 \times 10^5$ in 3 ml complete growth medium for 24 h and then co-transfected with a 160 μl 1:3 mixture of plasmids encoding GFP-tagged Scribble and Flag-tagged RPS6KA2 constructs of (3.3 μg DNA diluted in 155 μl Opti-MEM (Gibco) and 9.9 μl FuGENE HD (Promega)). For GST-pulldown, HEK293 cells were transfected with the 3xFlag CMV-10 RPS6KA2 plasmid or the mutant variants thereof or 3xFlag RPS6KA1 or mutants thereof. After 48 h, the cells were lysed with 1 ml lysis buffer (50 mM Tris–HCl, pH 7.4, 150 mM NaCl, 10 mM Na$_3$VO$_4$, 10 mM sodium pyrophosphate, 10 mM NaF, 1× cOmplete Mini EDTA-free protease inhibitor), supplemented with 0.5% Nonidet P-40, for 30 min at 4°C. 500 μl of the cell lysate was immunoprecipitated with anti-Flag M2 magnetic beads (Sigma-Aldrich) overnight at 4°C. One milliliter of the GST-pulldown lysates were incubated with 50 μg GST-Scribble PDZ1 for 1 h at 4°C and precipitated using GSH-magnetic beads (Sigma-Aldrich) overnight at 4°C. The beads were washed three times with 1 ml ice-cold lysis buffer followed by the addition of 50 μl SDS sample buffer (Bio-Rad) and incubation at 95°C for 10 min. The whole-cell lysate and the immunoprecipitate were separated on precast SDS–PAGE gels (Mini-PROTEAN TGX; Bio-Rad) and transferred to nitrocellulose membranes (Bio-Rad) for 16 h at 60 mA. The membranes were immunoblotted with anti-Flag (ab1162; Abcam), anti-GFP (ab6556; Abcam), and anti-actin (ab1801; Abcam) primary antibodies followed by HRP-conjugated anti rabbit secondary antibody (Life technologies) and detected with Amersham ECL Western Blotting Detection Reagent (GE Healthcare), and exposed to high-performance chemiluminescence film (Amersham Hyperfilm ECL; GE Healthcare) for 30–180 s and developed and fixed using GBX developer and fixer (Carestream Dental).

## Enrichment of predicted kinase phosphorylation sites

In order to assess kinase preferences for targets of the individual PDZ domains, as well as for Scribble and DLG1 targets as a whole, we employed NetworKIN (Linding *et al*, 2007) available in the KinomeXplorer online resource (Horn *et al*, 2014). To facilitate the prediction, we mapped the C-terminal sequences to the proteins in the String protein database (version 7.1 as used by NetworKIN) and removed instances for which no unique mapping was possible. We retrieved a maximum of 10 kinase substrate predictions per entry, using the standard parameters in the high-throughput workflow of the KinomeXplorer (minimum score 2 and maximum difference 4). We compared the frequency of kinase predictions between the background set (all mapped instances in the designed library) against predictions for individual binders of each PDZ domain and for all PDZ domains of Scribble or DLG1, respectively. Significance has been assessed using Fisher's exact test (one-sided), and *P*-values have been adjusted for multiple testing using the Bonferroni–Holmes correction.

## Overlap with literature and functional enrichments

The previously reported Scribble and DLG1 ligands were retrieved from Human Integrated Protein-Protein Interaction rEference (HIPPIE) on May 17, 2018 (Alanis-Lobato *et al*, 2017). Venn diagrams between known ligands and ligands found using phosphomimetic ProP-PD were made (http://bioinformatics.psb.ugent.be/

webtools/Venn/). The overlaps with our previously reported ProP-PD (Ivarsson *et al*, 2014) results were curated manually.

We used the two-set (set and background) comparison function of GOrilla (Eden *et al*, 2009) to compute functional enrichments of the selected binders versus the library sequences (with *P*-value cutoff $10^{-3}$). We found it necessary to use the actual library design as background, since a comparison of the library design to the complete human proteome revealed that the library itself is biased toward certain GO categories (see Table EV7). In concordance with the kinase enrichment analysis, we mapped the peptides to Ensembl identifiers and kept entries with unique mapping for the analysis.

### Network analysis

We created a network representation of the detected binders using Cytoscape (Shannon *et al*, 2003). In order to present the binders in a functional network context, we used the STRING database (Szklarczyk *et al*, 2015) and selected additional proteins which have a high-confidence (STRING score ≥ 0.8) connection to any of our bait or binder proteins. We further filtered for proteins within KEGG pathways selected from a STRING-based enrichment analysis (using the STRING web-page analysis section). The final network is presented in Fig 4C.

## Data availability

The data sets and computer code produced in this study are available in the following databases:

- Protein–protein interactions: The International Molecular Exchange Consortium IM-26482 (https://www.ebi.ac.uk/intact/query/IM-26482)
- NMR structure of Scribble PDZ1: Protein Data Bank (PDB) 6ESP (https://www.rcsb.org/structure/6ESP)

**Expanded View** for this article is available online.

### Acknowledgements
The phagemid used for library construction was generously provided by the Sidhu Lab (University of Toronto), together with expression constructs encoding GST-tagged proteins. The plasmid encoding GFP-tagged full-length human Scribble was generously provided by JP Borg (INSERM, Marseille). Access to the Monolith NT automated MST was provided by the SciLifeLab Drug Discovery and Development platform with initial technical guidance provided by Dr. Annette Roos. Sequencing was performed by the SNP&SEQ Technology Platform in Uppsala. The facility is part of the National Genomic Infrastructure (NGI) Sweden and Science for Life Laboratory. The SNP&SEQ Platform is also supported by the Swedish Research Council and the Knut and Alice Wallenberg Foundation. This work was supported by grants from the Swedish Research Council (YI, 2012-05092 and 2016-04965), the Wiberg Foundation (YI), and the Carl Trygger Foundation (YI, CTS15:226), and CNC was supported by a Wenner-Gren Foundation return fellow program starting grant.

### Author contributions

GNS, RA, CNC, and YI conceived experiments. RA designed the phage library, and GNS purified proteins, created the phage library, and performed biophysical and cell-based validations. CNC determined the NMR structure. MA performed affinity determinations. JO performed peptide NMR titrations. PG provided the CYANA program and participated in structure calculation. PN performed network analysis. GNS, RA, CNC, and YI wrote the manuscript.

### Conflict of interest

The authors declare that they have no conflict of interest.

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
