## [Review Process File · Molecular Systems Biology]

Proteome-wide analysis of phospho-regulated PDZ domain interactions

Gustav N. Sundell, Roland Arnold, Muhammad Ali, Piangfan Naksukpaiboon, Julien Orts, Peter Güntert, Celestine N. Chi & Ylva Ivarsson.

Review timeline:

Submission date:	29 th November 2017
Editorial Decision:	14 th February 2018
Revision received:	21 st June 2018
Editorial Decision:	11 th July 2018
Revision received:	24 th July 2018
Accepted:	24 th July 2018

Editor: Maria Polychronidou.

Transaction Report:

1st Editorial Decision

14th February 2018

Thank you again for submitting your work to Molecular Systems Biology. We have now heard back from the three referees who agreed to evaluate your study. As you will see below, the reviewers raise a series of concerns, which we would ask you to address in a major revision of the manuscript.

Without repeating all the points listed below, the referees recommended follow up analyses in order to increase the overall level of biological insight. Reviewer #3 makes constructive suggestions in this direction. Some of the more fundamental issues that need to be addressed are the following:

- As reviewer #3 points out, additional experimental evidence should be provided to better support that the results obtained using phosphomimetic peptides are relevant in a real biological setup.
- Statistical support for the presented findings should be provided where applicable, as mentioned by reviewer #2.

Of course all other issues raised by the referees would need to be thoroughly addressed.

REFeree REPORTS.

Reviewer #1:

This paper presents an approach for testing phosphomimetic (Ser/Thr - Glu) peptides in the context of peptide protein interactions. The authors developed a phage display coupled protein columns to rapidly identify peptides binding to target proteins. The approach was tested using a series of type I PDZ domains, that have previously been shown to be blocked in binding by phosphorylation at p-2,

but the authors explored a range of other potential phosphosites in several thousand PDZ-potential binders within the human proteome.

The results are several new interactions with the selected PDZ domains in addition to some surprising enabling interactions (ie. Where phosphorylation, for instance, and p-3 leads to enhanced or slightly enhanced binding). The authors do a good job of confirming several discoveries by ITC and even by solving an NMR structure. This additionally provides some mechanistic details for how the (e.g.) the p-3 phosphorylation might enhance binding.

Overall, I think this is a well presented study, and an approach that could readily be applied to several other instances, such as WW, 14-3-3 or other phosphoser/thr binding proteins. In general, this subject is rather underexplored and these kind of precise mechanistic details are absent for many important protein families. The systems where it can be applied to are, admittedly, somewhat limited, though perhaps even mimetics (Met/Gln) of Lysine acetylation might be used in some other approach, or more advanced genetics techniques used to exploit non-natural amino acids (e.g. better phosphomimetics or phospho-Tyrosine).

Comments:

Regarding my last point above, the authors didn't, as far as I could see, discuss much about phosphomimetic difficulties of Glu, notably the famous case of YWAZ 14-3-3 domain that does not bind to a Glu phosphomimetic (14578935); the use of non-natural amino acids might help here, and I wondered, momentarily if this might be possible within a phage-display system (sorry, I realise this might be naïve), but some discussion on the limitation of this might be in order.

It wasn't clear where the kinase information was coming from. I couldn't really read Table S1 that well (quality of the download). Nevertheless, I couldn't also ascertain what "predicted" meant.

I also wasn't sure if the phosphosites themselves (i.e. in the peptides) were known to be phosphorylated, or if the kinase was just known to phosphorylate the protein somewhere. Moreover, some insights into the known phosphorylation status (e.g. Phosphosite Plus, PHOSIDA, UniProt etc.) of the observed mimetic sites would help understanding and also possibly provide some additional evidence - apologies if this is already in a Table that I couldn't see.

For some of the determined motifs, there appears to be a mixture of E/D/T/S observed at specific positions (e.g. PDZ1 in Scribble position p-3; Scribble PDZ2, p-3; DLG1 PDZ1 p-2), suggesting possibly that constitutive phosphomimetics might exist in some peptides that are potentially phosphorylatable in other peptides (e.g. see Pearlmann et al., Ferrell Jr, Cell 2011). It often makes sense in these contexts to consider these together. It might even be fruitful to test some of these peptides as an additional validation (i.e. if one mimetic binds in addition to one natural Glu/Asp within a group).

Typos/minor comments:

Figure 5 a/b - some indication of where the peptide binding site is predicted to be would help relating the structure to the text (as in the supplementary figure).

Bottom of page 5, sentence garbled: "The library design contains 24% of all C-terminal non-redundant human C-terminal peptides of human full-length proteins reported in the annotated..."

Reviewer #2:

Report on Sundell et al. „Proteome-wide analysis of phospho-regulated PDZ domain interactions". Sundell et al. describe a peptide-phage display approach to assay binding sequences with PDZ domains. They use all human C-terminal sequences containing a S or T and replace these amino acid by E, as phosphomimetic, Six (rather 7, as another one is used in figure 6) PDZ domains are used

for peptide enrichment from the pool of 7600 sequences. After sequencing of the phage, binding motifs are revealed (S/T(-2)-x-Φ(0)) from a view sequences (6-20) that also contain E as binding variants at -3 (Scribble) and -8 (DGL1) for PDZ domains. Putative phospho-binding "motifs" do depend on as little as one peptide. These data are presented in much detail, including a biophysical characterization (MST) of the binding of selected peptides (3) and phospho-peptides. Data suggest phospho-specific binding. NMR experiments are used to - in a largely confirmatory manner with respect to the literature - define residues required for phospho-binding on the PDZ domain. Finally the SHANK1 PDZ domain is again used to define phospho-binding peptides, as the SHANK1 domain also contains the same critical residues for P-dependent binding. In summary the manuscript presents "proof of principle" [the authors] experiments to show that phospho-dependent peptide sequences can be selected using phosphomimetic substitutions in a phage display assay.

Some specific points for consideration.

- *) In general there is very little results in the manuscript and no biological insight. It is an in vitro study of some phospho / mimetic peptides binding to isolated PDZ domains (7 out of >250). The data are extensively presented, so that e.g. the assaying of the 7 th domain comes as a new figure 6. However it is the exact same experiment that was carried out at the start. This is some sort of story telling exercise, maybe. No statistical assessment of the data.
- *) Figure 2: Venn diagrams are misleading as the hippie PorDB suggest there is overlap of independent data, however the HIPPIE just incorporated the data.
- *) Figure 3b: the color code does not represent the data in a reasonably manner. The scale is from 0-1 there is a blue-red switch used that is not justified (whether there is one read more in wt or the mimic decides on the color). Some of the color code stems from one peptide etc ...
- *) It is very unclear why there are raw reads used and the input frequency in the library is not considered. This can potentially lead to strong enrichment of sequencing that are not uniformly represented in the original library. In general I do not know of any 2nd gen seq data that takes raw reads after selection for quantification without considering input. There is no significance measure (no data modeling) as to whether the blue or the red bar may be higher or lower (Figure 3a).
- *) No bioinformatics methods part. How have the PWMs generated, what is the background, the significance?
- *) The different effects of phospho-mimic peptides and phosphorylated peptides remain unexplained.

Reviewer #3:

The study is interesting. It provides a new method and a workflow of methods to provide mechanistic insight into modular domain binding in vitro. Its clear the study has no cell/in vivo data but that is also not the point of the study, its a strictly biophysical study so to speak. That being said it needs to be quite significantly improved before publication:

1. The authors provide limited and quite naive bioinformatics analysis:

- * There is little discussion of how the wild-type, E substituted (mimetic) and known motifs overlap and are separate in the data/experiments. Eg. do the mimetic binding compete with wt-phospho peptides? this could be an interesting control experiment.
- * Would it be possible to better integrate the data set with known interactions at the network level?
- * The section on kinases is under developed, how does the data set compare with motifs/predictions from eg. KinomXplorer

2. The main issue is to somehow convince that looking at mimetics are representing what one would get looking at the wt situation (ie. phosphorylated peptides).

- * Could the authors not phosphorylate the library of control peptides to compare to. Eg. at least for one of the domains order the wt set (~4827) in phosphorylated form. This set should then be put through the same binding experiment as with 1-2 of the domains tested?
- * The motifs from this could then be compared to the mimetic ones to see if there are differences/similarities.

* Mimetics is something that has mainly been observed for kinase activation loops (BRAF V600E) where they create an allosteric re-arrangement resulting in constitutive active kinase. It is not so clear how this concept translate into peptide binding experiments.

3. The authors should perhaps make the last point above the main investigation topic of the paper. I.e. try to understand to which degree the concept of phospho-mimetic 'translates' from kinases to PDZ binding. They should analyze and present this more direct and make it the main focus for the discussion. It is ok if the concept does not translate or if it only does so with limited success; then we learnt something new!

It appears to the reviewer as that the authors would benefit from having some more direct discussions with other peers in the field to find new ways to analyze and understand the data obtained, as such the study would also benefit from a specific mode of publication such as 'resource' or perhaps accompanied by a website that enables the reader to explore the data in more details and comparison to other 'views' of the PDZ binding landscape for Scribble and DLG1.

1st Revision - authors' response

21st June 2018

Point by point answer to the reviewers' comments:

Reviewer 1 (R#1)

R#1 comment 1:

The authors didn't, as far as I could see, discuss much about phosphomimetic difficulties of Glu, notably the famous case of YWAZ 14-3-3 domain that does not bind to a Glu phosphomimetic (14578935); the use of non-natural amino acids might help here, and I wondered, momentarily if this might be possible within a phage-display system (sorry, I realise this might be naïve), but some discussion on the limitation of this might be in order.

Reply:

We have added an extensive discussion on the pros and cons of using phosphomimetic mutations. We have also included a discussion about the use of non-natural amino acids in phage display.

“The rationale behind using Glu as a mimetic mutation of pSer/pThr is that Glu (or Asp) is negatively charged and is nearly isosteric with pSer and pThr (Pearlman et al, 2011). Phosphomimetic mutations have frequently been used to study the effects of phosphorylation since it was shown that replacing a phosphoserine with an Asp mimicked the phosphorylated state of a protein (Thorsness & Koshland, 1987). It also appears as if nature have made use of the reverse engineering by evolving phosphosites from acidic amino acids (Pearlman et al, 2011). It has, however, been less common to use phosphomimetic mutations to study the effects of phosphorylation on a ligand binding site. The success of the approach will depend on the nature of the interactions studied. On one side, phosphomimetic mutations serve a qualitative purpose when the intention is to investigate a general charge dependent interaction although the quantitative effects are not expected to be identical given the differences in net charge at physiological pH (Glu -1, pSer/pThr -2) (Cooper et al, 1983) and the differences in hydration shells (Hunter, 2012). Along these lines, we found the effects of phosphomimetic mutations and ligand phosphorylation on PDZ binding to be consistently enabling or disabling depending on the ligand position, although the absolute numbers differed. Similar results were reported in a recent study on the effect of p-2 phosphorylation of the EQVSAV-coo- peptide of on binding to the second PDZ domain of PTBL (Toto et al, 2017). In a similar fashion, acidic amino acids or with phospho-Ser/The downstream of the core docking motif of the regulatory subunit B56 of protein phosphatase 2A (L.I.E) enhance the affinity of the interaction, as also shown by ProP-PD (Hertz et al, 2016; Wu et al, 2017). In contrast, phosphomimetic mutations are less suitable for mimicking Ser phosphorylation the second position of the motif, as it involves specific interactions with an Arg that is too far away to be reached by a phosphomimetic mutation (Wang et al, 2016). The phosphomimetic ProP-PD approach should thus be applicable to other cases, for which negative charges contributes to the binding affinity but where there is no strict requirement on phosphorylation, and also to explore disabling effects of phosphomimetic mutations. For example, the WD40 repeats are among the most abundant protein domains and they form versatile interaction modules. Phosphomimetic mutations have been found to

enable interactions with WD40-repeat of UAF1 (Villamil et al, 2012) and β TrCP (da Silva Almeida et al, 2012), and could represent suitable bait proteins for phosphomimetic ProP-PD.

On the other side, when it comes to exploring a specific phospho-binding site, there is often a strong dependence of the geometry of the Args (or other residues) that line the binding pocket. For example, the FHA domain of Rad53p requires the presence of phospho-Thr and does not tolerate phosphoserine or Glu (Durocher et al, 1999). For such cases, phosphomimetic mutations are bound to fail. Another example for which a phosphomimetic mutation has been shown to fail to have the desired effect involves 14-3-3 ζ (Zheng et al, 2003). However, in a recent study it was found that a phosphomimetic RxEP motif in the NS3 protein of dengue virus interacts with 14-3-3 ϵ (Chan & Gack, 2016). Given the versatility of the binding mode of the 14-3-3 proteins it is possible that the results will depend on the overall context of the peptide subjected to phosphomimetic mutation. To explore the full potential of phosphomimetic ProP-PD it will be necessary to design follow-up studies with more complex ProP-PD libraries (e.g. not limited to C-terminal regions and a panel of phospho-peptide binding domains).

A more sophisticated approach to tackle PTM dependent interactions would be to use an expanded genetic code, where an amber stop codon is used to encode the 21 amino acid, in combination with phage display experiments. Given that an expanded genetic code has been used for protein evolution experiments with several unnatural amino acids (Liu et al, 2008) and that optimized orthogonal aminoacyl-tRNA synthetase/tRNACUA pairs for phospho-Ser and phospho-Thr incorporation were recently generated (Rogerson et al, 2015; Zhang et al, 2017), such an approach should be feasible, and is likely to become a viable approach in specialized laboratories. For such a set-up, it will be crucial to ensure that the full-length phosphoserine containing peptides are produced and displayed at the same level as other peptides, and that they do not suffer any growth disadvantage. Unless such biases are mitigated, the results will be misleading. On a related note, a recent study encoded human phosphoserine containing in bacteria for profiling of phosphodependent interactions. Thus, this and other emerging approaches will be of complementary use for finding phosphorylation-dependent interactions (Barber et al, 2018). Among them, phosphomimetic ProP-PD offers an efficient approach for detecting potential phosphorylation switches without the requirement of the usage of alternative codons or optimization of protocols and thus offers a straightforward and attractive approach of broad applicability for exploring the potential phospho-switching of motif-based interactions.”

R#1 Comment 2:

It wasn't clear where the kinase information was coming from. I couldn't really read Table S1 that well (quality of the download). Nevertheless, I couldn't also ascertain what "predicted" meant. I also wasn't sure if the phosphosites themselves (i.e. in the peptides) were known to be phosphorylated, or if the kinase was just known to phosphorylate the protein somewhere. Moreover, some insights into the known phosphorylation status (e.g. Phosphosite Plus, PHOSIDA, UniProt etc.) of the observed mimetic sites would help understanding and also possibly provide some additional evidence - apologies if this is already in a Table that I couldn't see.

Reply:

The method part of the library design was accidentally omitted from the method section. The following section has now been added to the Methods:

“We selected human C-terminal sequences (9mers) containing known Ser or Thr phosphosites from Phosphosite 2,57, Phospho-ELM58 as downloaded on the 10.10. 2014, and Netphos 3.159 predictions on C-terminal wild-type sequences from human Swissprot/Uniprot (as downloaded from the 10.10 2014) and created a phage library design that contains all these sequences as wild-type and phosphomimetic instances. We also added combinatorial entries in case of multiple predicted or measured sites: if, for example, two phosphorylation sites (A,B) were found in the same sequence, we added pA+Bw, Aw+pB, and pA+pB (where ‘p’ denotes the phosphomimetic variant, and w the wildtype). This resulted in a total of 12455 entries in the design (Table EV1). In case of Phospho-ELM and the Netphos predictions, we provide the information on the specific kinase in the library design file. To generate the oligonucleotide sequences required for library construction, peptide sequences were reversely translated and flanked by annealing sites complementary to the phagemid vector (Table EV2).”

The requested information was nevertheless provided in the supplemental table 1 (library design). We apologies for the quality of the table. The reason for the poor quality was due to the fact that it is

an extensive file that is better provided as an excel file. Unfortunately, there was as far as we could see no option of uploading an excel file during the initial submission, so the table became compressed into a PDF file.

To clarify, sequences were added to the design, if there was either direct evidences from databases/experiments (as collected from PhosphoElm, and PhosphoSite), or predicted evidence by NetPhos supporting that they may be subjected to Ser/Thr phosphorylation. The phosphosites are in the peptides as indicated in the former Sup Table 1, current Table EV1).

R#1 Comment 3:

For some of the determined motifs, there appears to be a mixture of E/D/T/S observed at specific positions (e.g. PDZ1 in Scribble position p-3; Scribble PDZ2, p-3; DLG1 PDZ1 p-2), suggesting possibly that constitutive phosphomimetics might exist in some peptides that are potentially phosphorylatable in other peptides (e.g. see Pearlmann et al., Ferrell Jr, Cell 2011). It often makes sense in these contexts to consider these together. It might even be fruitful to test some of these peptides as an additional validation (i.e. if one mimetic binds in addition to one natural Glu/Asp within a group).

Reply:

Indeed, for several cases there are acidic residues on positions that were pinpointed to be preferentially occupied by phosphomimetic mutations over Ser/Thr, consistent with the preferences for acidic residues on these positions. We found that determining affinities for peptides with naturally occurring E or D on a proposed regulatory site out of the intended scope of the paper. However, we analyzed the conservation of ligands with known or putative phospho-sites and found them to be generally well conserved, and there were as far as we could see no naturally occurring phosphomimetics of the suggested phospho-sites within metazoan.

Typos/minor comments:

R#1 Comment 4:

Figure 5 a/b - some indication of where the peptide binding site is predicted to be would help relating the structure to the text (as in the supplementary figure).

Reply:

We now indicate the canonical peptide binding site in the structure as suggested by the reviewer.

R#1 Comment 5:

Bottom of page 5, sentence garbled: "The library design contains 24% of all C-terminal non-redundant human C-terminal peptides of human full-length proteins reported in the annotated..."

Reply:

We have corrected the text.

Reviewer #2:

Report on Sundell et al., "Proteome-wide analysis of phospho-regulated PDZ domain interactions".

Sundell et al. describe a peptide-phage display approach to assay binding sequences with PDZ domains. They use all human C-terminal sequences containing a S or T and replace these amino acid by E, as phosphomimetic. Six (rather 7, as another one is used in figure 6) PDZ domains are used for peptide enrichment from the pool of 7600 sequences. After sequencing of the phage, binding motifs are revealed (S/T(-2)-x-Φ(0)) from a view sequences (6-20) that also contain E as binding variants at -3 (Scribble) and -8 (DGL1) for PDZ domains. Putative phospho-binding "motifs" do depend on as little as one peptide. These data are presented in much detail, including a biophysical characterization (MST) of the binding of selected peptides (3) and phospho-peptides. Data suggest phospho-specific binding. NMR experiments are used to - in a largely confirmatory manner with respect to the literature - define residues required for phospho-binding on the PDZ domain. Finally the SHANK1 PDZ domain is again used to define phospho-binding peptides, as the SHANK1

domain also contains the same critical residues for P-dependent binding.

In summary the manuscript presents "proof of principle" [the authors] experiments to show that phospho-dependent peptide sequences can be selected using phosphomimetic substitutions in a phage display assay.

Some specific points for consideration.

R#2 Comment 1:

*) In general there is very little results in the manuscript and no biological insight. It is an in vitro study of some phospho / mimetic peptides binding to isolated PDZ domains (7 out of >250). The data are extensively presented, so that e.g. the assaying of the 7th domain comes as a new figure 6. However it is the exact same experiment that was carried out at the start. This is some sort of story telling exercise, maybe. No statistical assessment of the data.

Reply:

The aim of the study was a proof-of-principle study of phosphomimetic ProP-PD, not to perform a comprehensive analysis of the phospho-peptide binding of the full PDZ domain family. This could be a perfectly valid aim for another study. Nevertheless, we have added qualitative results on phosphomimetic ProP-PD selections of additional PDZ domain (Fig 5; Table EV9).

The analysis of the 7th domain in the initial manuscript was conducted towards the end of the story when the method had been validated and we reflected upon available information on phosphopeptide binding of other PDZ domains. The PDZ domain of Shank had previously been suggested to bind phosphopeptides. Surprisingly, we found that that was not the case when tested by phosphomimetic ProP-PD or through affinity measurements, and we included it as we realized that it provided an added value to the manuscript.

R#2 Comment 2:

Figure 2: Venn diagrams are misleading as the hippie PorDB suggest there is overlap of independent data, however the HIPPIE just incorporated the data.

Reply:

We aimed to clarify that a part of the information in the Hippie database that overlaps with our new results stems from our previous ProP-PD data that has been incorporated into the databases of protein-protein interactions (e.g. IntAct) and therefore appears in the Hippie database. Thus, the intention was the opposite of how the reviewer interpreted it. To avoid any confusion, we have now updated the layout of the figure (see Figure 4).

R#2 Comment 3:

Figure 3b: the color code does not represent the data in a reasonable manner. The scale is from 0-1 there is a blue-red switch used that is not justified (whether there is one read more in wt or the mimic decides on the color). Some of the color code stems from one peptide etc ...

Reply:

To represent the data in a better way we have include bar graphs of the NGS results that clearly indicates the fraction of counts the peptides obtained in the NGS analysis of the binding enriched phage pools.

Following the reviewer's suggestion, we further changed the gradient used in the matrices from red-white-blue to red-yellow-blue as it gives a clearer color difference for the scores. We have summarized the statistics of the data in Table EV4.

R#2 Comment 4:

It is very unclear why there are raw reads used and the input frequency in the library is not considered. This can potentially lead to strong enrichment of sequencing that are not uniformly represented in the original library. In general I do not know of any 2nd gen seq data that takes raw reads after selection for quantification without considering input.

Reply:

During phage display experiments with repeated rounds of panning there is a strong selection for binding ligands such that it is unlikely that minor biases in the naïve library used as input day 1 is reflected in any major bias in the NGS analysis of the binding enriched phage pools after the third round of selection. In this case, we have sequenced the libraries to 92% or 96% coverages. We found no significant correlation of the actual read counts (selection ‘strength’) and the initial representation in the library, reflecting the strong selection during phage display. There is no major bias in the input library composition, and it is not reflected in the final counts. Consequently, no modelling of the input library frequency to score the interactions is necessary or feasible. We have added a section on this analysis to the method sections, as well as a Fig. EV1 to clarify the issue.

R#2 Comment 5:

There is no significance measure (no data modeling) as to whether the blue or the red bar may be higher or lower (Figure 3a).

Reply:

We have indicated the cases for which the differences between the wild-type and phosphomimetic peptides are significant.

R#2 Comment 6:

No bioinformatics methods part. How have the PWMs generated, what is the background, the significance?

Reply:

The method part on the library part had accidentally been removed, it has now been added. Regarding the PWMs, they were generated using WebLogo, after aligning the peptides based on the C-termini (as PDZ domains bind C-terminal peptides). The background is flat (see below for PWM generated based on the library design).

R#2 Comment 7:

The different effects of phospho-mimic peptides and phosphorylated peptides remain unexplained.

Reply:

We have added an extensive discussion on the pros and cons with using phosphomimetic mutations that clarifies this question.

Reviewer #3:

The study is interesting. It provides a new method and a workflow of methods to provide mechanistic insight into modular domain binding *in vitro*. It's clear the study has no cell/*in vivo* data but that is also not the point of the study, it's a strictly biophysical study so to speak. That being said it needs to be quite significantly improved before publication.

Reply:

Thanks to the helpful suggestions of the reviewers we have significantly improved the manuscript. We have added novel experimental validations (pre-phosphorylation ProP-PD, new Fig. 6), developed the discussion and added statistical analysis.

R#3 Comment 1:

There is little discussion of how the wild-type, E substituted (mimetic) and known motifs overlap and are separate in the data/experiments. Eg. do the mimetic binding compete with wt-phospho peptides? this could be an interesting control experiment.

Reply:

As shown by the NMR structure the unphosphorylated and phosphorylated ligands (as well as the phosphomimetic peptides) compete the same binding pocket, that is, the canonical PDZ peptide binding groove. Thus, the ligands compete with each other and we judge that that this is an important clarification to this point.

R#3 Comment 2:

Would it be possible to better integrate the data set with known interactions at the network level?

Reply:

Following the suggestion of the reviewer we have integrated the novel interactions into an extended network, see Fig. 4.

R#3 Comment 3:

The section on kinases is under developed, how does the data set compare with motifs/predictions from eg. KinomXplorer

Reply:

We agree. We performed the analysis using KinomXplorer and found that the results were largely the same. We have replaced the previous analysis that was based on the kinase information in the library design, with an a NetworkIn analysis. Analysis.

R#3 Comment 4:

The main issue is to somehow convince that looking at mimetics are representing what one would get looking at the wt situation (ie. phosphorylated peptides). Could the authors not phosphorylate the library of control peptides to compare to. Eg. at least for one of the domains order the wt set (~4827) in phosphorylated form. This set should then be put through the same binding experiment as with 1-2 of the domains tested? The motifs from this could then be compared to the mimetic ones to see if there are differences/similarities.

Reply:

Following the constructive suggestions of the reviewer we generated a library of control peptides, established a protocol for phosphorylating it using RPS6KA1 and used it in unphosphorylated or pre-phosphorylated form in selections against Scribble PDZ1 (see Fig 6).

A significant difference between the selections in absence and presence of the kinase is a 3 times higher frequency of the TMFLRETSL-coo- peptide of GUYCA2 in the NGS results of the binding enriched phage pools from the selection against the phosphorylated library. This peptide contains a RxxS motif, which represent a typical phosphorylation site of RPS6KA1. Thus, the experiment supports that p-1 phosphorylation enables Scribble PDZ1 interactions, and that the results generated in the study may be of biological relevance. We find that this is an interesting approach that can be used in tandem with the phosphomimetic ProP-PD.

The experiment is however somewhat difficult to control (e.g. we don't know the phosphorylation degree) and is biased towards phospho-sites towards the more C-terminal part of the peptide as the kinase preferentially acts on ligands with a Arg two positions upstream. Thus, additional method optimization would be necessary before using the protocol on a regular basis.

R#3 Comment 4 & 5

Mimetics is something that has mainly been observed for kinase activation loops (BRAF V600E) where they create a allosteric re-arrangement resulting in constitutive active kinase. It is not so clear how this concept translate into peptide binding experiments.

The authors should perhaps make the last point above the main investigation topic of the paper. I.e. try to understand to which degree the concept of phospho-mimetic 'translates' from kinases to PDZ binding. They should analyze and present this more direct and make it the main focus for the discussion. It is ok if the concept does not translate or if it only does so with limited success; then we learnt something new!

Reply:

We have added an extensive discussion on the topic of the pros and cons of phosphomimetic mutations and peptide binding experiments and in put it the context of the original kinase phosphomimetic studies. We also discuss when the phosphomimetic ProP-PD approach may be applicable, and when it might not be a suitable approach.

R#3 Comment 6:

It appears to the reviewer as that the authors would benefit from having some more direct discussions with other peers in the field to find new ways to analyze and understand the data obtained, as such the study would also benefit from a specific mode of publication such as 'resource' or perhaps accompanied by a website that enables the reader to explore the data in more details and comparison to other 'views' of the PDZ binding landscape for Scribble and DLG1.

Reply:

We find that data set of the current study is too limited to merit a resource type publication. However, we agree that this data set, together with our growing data sets from other studies, should be made be more easily assessable than by searching supplemental files. We aim to generate a database that contain the growing information generated by proteomic phage display in the near future.

2nd Editorial Decision

11th July 2018

Thank you for sending us your revised manuscript. We have now heard back from reviewer #3 who was asked to evaluate your study. As you will see below, s/he is satisfied with the performed revisions and is supportive of publication.

Before formally accepting your manuscript for publication, we would ask you to address the following remaining editorial issues:

REFeree REPORTS.

Reviewer #3:

The authors have answered my comments.

The paper should be scheduled for publication.